# Coil-to-α-helix transition at the Nup358-BicD2 interface activates BicD2 for dynein recruitment

**James M Gibson**[1†], **Heying Cui**[2†], **M Yusuf Ali**[3†], **Xiaoxin Zhao**[2], **Erik W Debler**[4‡], **Jing Zhao**[1§], **Kathleen M Trybus**[3*], **Sozanne R Solmaz**[2*], **Chunyu Wang**[1*]

[1]Department of Biological Sciences, Department of Chemistry and Chemical Biology, Center for Biotechnology and Interdisciplinary Studies, Rensselaer Polytechnic Institute, Troy, United States; [2]Department of Chemistry, Binghamton University, Binghamton, United States; [3]Department of Molecular Physiology and Biophysics, Larner College of Medicine, University of Vermont, Burlington, United States; [4]Laboratory of Cell Biology, The Rockefeller University, New York, United States

**\*For correspondence:**
Kathleen.Trybus@med.uvm.edu
(KMT);
ssolmaz@binghamton.edu (SRS);
wangc5@rpi.edu (CW)

[†]These authors contributed
equally to this work

**Present address:** [‡]Department
of Biochemistry & Molecular
Biology, Thomas Jefferson
University, Philadelphia, United
States; [§]College of Food Science
and Nutritional Engineering,
China Agricultural University,
Beijing, China

**Competing interest:** The authors
declare that no competing
interests exist.

**Reviewing Editor:** Andrew
P Carter, MRC Laboratory of
Molecular Biology, United
Kingdom

**Abstract** Nup358, a protein of the nuclear pore complex, facilitates a nuclear positioning pathway that is essential for many biological processes, including neuromuscular and brain development. Nup358 interacts with the dynein adaptor Bicaudal D2 (BicD2), which in turn recruits the dynein machinery to position the nucleus. However, the molecular mechanisms of the Nup358/BicD2 interaction and the activation of transport remain poorly understood. Here for the first time, we show that a minimal Nup358 domain activates dynein/dynactin/BicD2 for processive motility on microtubules. Using nuclear magnetic resonance titration and chemical exchange saturation transfer, mutagenesis, and circular dichroism spectroscopy, a Nup358 α-helix encompassing residues 2162–2184 was identified, which transitioned from a random coil to an α-helical conformation upon BicD2 binding and formed the core of the Nup358-BicD2 interface. Mutations in this region of Nup358 decreased the Nup358/BicD2 interaction, resulting in decreased dynein recruitment and impaired motility. BicD2 thus recognizes Nup358 through a 'cargo recognition α-helix,' a structural feature that may stabilize BicD2 in its activated state and promote processive dynein motility.

## Editor's evaluation

The paper provides evidence that a peptide from the nuclear pore protein Nup358 (RanBP2) forms a helix upon binding the dynein/dynactin adaptor BICD2 and is able to activate BICD in the same way as the small G-protein Rab6. Helix formation to activate BicD2 is a novel concept and will open up the search space for other potential activators that may be unstructured in their apo states.

## Introduction

Cytoplasmic dynein is the predominant motor responsible for minus-end-directed traffic on microtubules (*Reck-Peterson et al., 2018*), which facilitates a vast number of transport events that are critical for chromosome segregation, signal transmission at synapses, and brain and muscle development (*Fu and Holzbaur, 2014*; *Hendricks et al., 2010*; *Wilson and Holzbaur, 2012*; *Zhu et al., 2017*; *Splinter et al., 2010*; *Bolhy et al., 2011*; *Morris and Hollenbeck, 1993*; *Soppina et al., 2009*; *Encalada et al., 2011*; *Dharan and Campbell, 2018*; *Hu et al., 2013*; *Baffet et al., 2015*; *Gonçalves et al., 2020*; *Zhang et al., 2007*; *Zhang et al., 2009*). Integral to the transport machinery are *dynein adaptors*, such as Bicaudal D2 (BicD2), whose N-terminal region (BicD2[CC1]) recruits and

activates dynein-dynactin (DD) for processive motility (*Splinter et al., 2012*; *Schlager et al., 2014b*; *Schlager et al., 2014a*; *Hoogenraad et al., 2003*; *Hoogenraad et al., 2001*; *McKenney et al., 2014*; *Urnavicius et al., 2018*; *Urnavicius et al., 2015*; *Sladewski et al., 2018*; *McClintock et al., 2018*; *Liu et al., 2013*). Also integral to the dynein transport machinery are *cargoes*, which bind to the C-terminal domain (CTD) of BicD2 (*Matanis et al., 2002*). Cargoes are required to activate BicD2 for dynein binding, which is a key regulatory step for dynein-dependent transport (*Splinter et al., 2012*; *Schlager et al., 2014b*; *Schlager et al., 2014a*; *Hoogenraad et al., 2003*; *Hoogenraad et al., 2001*; *McKenney et al., 2014*; *Urnavicius et al., 2018*; *Urnavicius et al., 2015*; *Sladewski et al., 2018*; *McClintock et al., 2018*; *Liu et al., 2013*). In the absence of cargo adaptor/dynein adaptor complexes such as Nup358/BicD2, dynein and dynactin are autoinhibited and only show diffusive motion on microtubules. Furthermore, in the absence of cargoes, BicD2 assumes a looped, autoinhibited conformation, in which its N-terminal dynein/dynactin-binding site binds to the CTD and remains inaccessible. The CTD is required for autoinhibition as a truncated BicD2 without the CTD activates dynein/dynactin for processive motility. Binding of cargo to the CTD releases auto-inhibition, likely resulting in an extended conformation that recruits dynein and dynactin (*Splinter et al., 2012*; *Schlager et al., 2014b*; *Schlager et al., 2014a*; *Hoogenraad et al., 2003*; *Hoogenraad et al., 2001*; *McKenney et al., 2014*; *Urnavicius et al., 2018*; *Urnavicius et al., 2015*; *Sladewski et al., 2018*; *McClintock et al., 2018*; *Liu et al., 2013*). The crystal structures of the C-terminal cargo recognition domains of three BicD2 homologs have been determined (*Liu et al., 2013*; *Terawaki et al., 2015*; *Noell et al., 2019*). However, to date, there are no detailed structural studies of BicD2/cargo complexes and the structural mechanisms of BicD2-mediated cargo recognition and dynein activation remain poorly understood.

The binding partners for human BicD2 include Nup358 (*Splinter et al., 2010*), Rab6$^{GTP}$ (*Matanis et al., 2002*), and nesprin-2G (*Gonçalves et al., 2020*). Nup358, also known as RanBP2, is a 358 kDa nuclear pore complex protein with multiple functions (*Wu et al., 1995*). During G2 phase, Nup358 engages in a pathway for positioning of the nucleus relative to the centrosome along microtubules by binding to BicD2, which in turn recruits dynein and dynactin (*Figure 1*; *Splinter et al., 2010*). This pathway is essential for apical nuclear migration during differentiation of radial glial progenitor cells, which give rise to the majority of neurons and glia cells of the neocortex (*Hu et al., 2013*; *Baffet et al., 2015*). A second nuclear positioning pathway is facilitated by BicD2/dynein and nesprin-2G (*Gonçalves et al., 2020*), a component of linker of nucleoskeleton and cytoskeleton (LINC) complexes (*Fridolfsson et al., 2010*), which is important for migration of post-mitotic neurons during brain development (*Gonçalves et al., 2020*). Apart from its roles in nuclear positioning, BicD2 is also involved in the transport of Golgi-derived and secretory vesicles. In this process, BicD2 and dynein are recruited by the vesicle-associated small GTPase Rab6 (*Matanis et al., 2002*; *Grigoriev et al., 2007*). Thus, BicD2 plays important roles in faithful chromosome segregation, neurotransmission at synapses, as well as brain and muscle development (*Splinter et al., 2010*; *Hu et al., 2013*; *Baffet et al., 2015*; *Gonçalves et al., 2020*; *Zhang et al., 2009*; *Matanis et al., 2002*). Mutations in BicD2 cause neuro-muscular diseases, including a subset of spinal muscular atrophy cases (*Neveling et al., 2013*; *Peeters et al., 2013*; *Martinez-Carrera and Wirth, 2015*; *Tsai et al., 2020*; *Synofzik et al., 2014*), the most common genetic cause of infant death (*Meijboom et al., 2017*).

Within the Nup358 sequence, there are many intrinsically disordered regions (IDRs), which are also commonly found in proteins involved in dynein-dependent transport (*Zhu et al., 2017*; *Celestino et al., 2019*; *Lee et al., 2020*; *Henen et al., 2021*; *Lee et al., 2018*). Although IDRs and intrinsically disordered proteins (IDPs) make up ~30% of eukaryotic proteins and have important physiological functions (*Ward et al., 2004*), they remain the most poorly characterized class of proteins in regards to their structure, dynamics, and interactions. IDRs play important roles in dynein biology. For example, an IDR in the dynein light intermediate chain 1 (LIC1) undergoes a coil-to-α-helix transition when it interacts with the N-terminal domain of BicD2, which is an important step in the activation of dynein for processive motility (*Celestino et al., 2019*; *Lee et al., 2020*; *Lee et al., 2018*). A second, larger interface is formed between the N-terminal coiled coil of BicD2, the dynein tail, and dynactin, which promotes activation of dynein for processive motility (*Splinter et al., 2012*; *Schlager et al., 2014b*; *McKenney et al., 2014*; *Urnavicius et al., 2015*). Currently, the mechanism of dynein activation by full length BicD2/cargo complexes is not well understood since much of the available information pertaining to dynein activation was derived from studies with isolated BicD2 fragments.

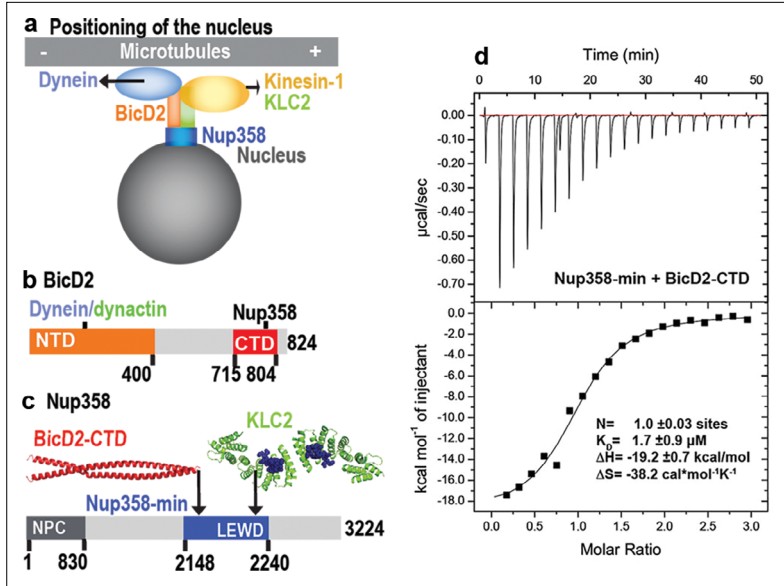

**Figure 1.** A minimal Nup358 domain interacts with BicD2 with micromolar affinity. (**a**) Nup358 interacts with both BicD2/dynein/dynactin and kinesin-1 (via kinesin-1 light chain 2, KLC2) to mediate bidirectional nuclear positioning in G2 phase of the cell cycle (*Splinter et al., 2010*; *Cai et al., 2001*; *Cui et al., 2019*). This pathway is essential for a fundamental process in brain development that is required for radial glial progenitor cells to differentiate to the majority of neurons and glia cells of the neocortex (*Hu et al., 2013*). (**b, c**) Schematic representation of the expression constructs BicD2-CTD (**b**, red) and Nup358-min (**c**, blue), in the context of the full-length proteins (gray). (**c**) KLC2 is recruited to Nup358-min via a W-acidic motif with the sequence LEWD (*Cui et al., 2019*). The X-ray structures of the TPR domain of the KLC2 (green, referred to as KLC2 hereafter), fused to a LEWD motif (purple) (*Pernigo et al., 2013*), and the X-ray structure of the BicD2-CTD (red) (*Noell et al., 2019*) are shown in cartoon representation in (**c**). The α-helical N-terminal domain of Nup358 (dark gray) promotes anchorage to the nuclear pore complex (NPC). (**d**) A representative isothermal titration calorimetry (ITC) thermogram of Nup358-min and BicD2-CTD is shown, from which the affinity (dissociation constant $K_D$) was determined to be 1.7 ± 0.9 μM. N: number of sites; ΔS: the change in entropy; ΔH: the change in enthalpy. The experiment was repeated three times. The error of $K_D$ was calculated as the standard deviation from three experiments. In *Figure 1—figure supplement 1*, the affinity of Nup358-min-GST (i.e., with the GST-tag intact) and BicD2-CTD was determined by ITC to be 1.6 ± 1.0 μM. See also *Figure 1—source data 1–4*.

The online version of this article includes the following source data and figure supplement(s) for figure 1:

**Source data 1.** ITC thermogram of BicD2-CTD and Nup358-min.

**Source data 2.** ITC thermogram of BicD2-CTD and Nup358-min-GST (i.e., with the GST-tag intact).

**Source data 3.** ITC thermogram of Nup358-min into buffer.

**Source data 4.** ITC thermogram of Nup358-min-GST into buffer.

**Figure supplement 1.** The GST-fusion has a neglible effect on the interaction between BicD2-CTD and Nup358-min.

---

In addition to BicD2, Nup358 also recruits the opposite polarity motor kinesin-1 via the subunit kinesin-1 light chain 2 (KLC2), which binds to a W-acidic motif with the sequence LEWD in Nup358 (*Cai et al., 2001*; *Cui et al., 2019*; *Dodding et al., 2011*). While dynein is the predominant motor in G2 phase, kinesin-1 is also actively involved in nuclear positioning in G2 phase, modulating overall motility (*Splinter et al., 2010*). Such bidirectional transport is also displayed by mitochondria, endosomes, viruses, phagosomes, secretory vesicles, and many vesicles in neuronal axons and growth cones (*Fu and Holzbaur, 2014*; *Hendricks et al., 2010*; *Wilson and Holzbaur, 2012*; *Zhu et al., 2017*; *Splinter et al., 2010*; *Bolhy et al., 2011*; *Morris and Hollenbeck, 1993*; *Soppina et al., 2009*; *Encalada et al., 2011*; *Dharan and Campbell, 2018*; *Gonçalves et al., 2020*). Opposite polarity motors such as BicD2/dynein and kinesin-1 often bind in close spatial proximity to adapter-binding proteins such as Nup358, but it is unknown how their overall motility is regulated, likely because their interactions remain poorly characterized by structural and biophysical methods.

Here, we have determined the structural properties of the interface of a minimal Nup358/BicD2 complex by a combination of nuclear magnetic resonance (NMR) spectroscopy, mutagenesis, circular dichroism (CD) spectroscopy, and small-angle X-ray scattering. These results establish a structural basis for cargo recognition by BicD2 and suggest that Nup358 interacts with BicD2 through a 'cargo recognition' α-helix. Direct activation of BicD2 by Nup358 has not been shown before. Here, we use single-molecule-binding and processivity assays to show that a minimal dimerized Nup358 construct is sufficient to activate full-length dynein/dynactin/BicD2 complexes for processive motility. Mutations of the cargo recognition α-helix of Nup358 decreased the Nup358/BicD2 interaction and resulted in decreased dynein recruitment and impaired motility, shedding light on the important role of the cargo recognition α-helix for the activation of BicD2 and dynein motility. Intriguingly, our results also show that the binding site of BicD2 in Nup358 is spatially close to but does not overlap with the LEWD motif that acts as a kinesin-1-binding site, suggesting that the kinesin and dynein machineries may interact simultaneously via Nup358. Our results thus provide mechanistic insights into the regulation of bidirectional transport by adapter-binding proteins.

## Results

### ITC establishes a minimal complex for Nup358/BicD2 interaction

Previously, we have determined an X-ray structure of the CTD of human BicD2 (BicD2-CTD, residues 715–804), which contains the binding sites for cargoes, including human Nup358 (*Figure 1b*; *Noell et al., 2019*). A complex was reconstituted with BicD2-CTD and a minimal fragment of human Nup358 containing residues 2148–2240, which is called Nup358-min (*Noell et al., 2019*; *Cui et al., 2019*; *Figure 1c*). Here, the affinity of the BicD2-CTD towards Nup358-min was determined by isothermal titration calorimetry (ITC) (*Figure 1d*, *Figure 1—figure supplement 1*). The ITC thermogram fits well to a one-site-binding model (with a single-equilibrium dissociation constant $K_D$). The number of sites was determined to be N = 1.0, which is consistent with a molar ratio of [Nup358]/[BicD2] of 1. This molar ratio is in agreement with our previously published molar masses obtained from size-exclusion chromatography coupled to multi-angle light scattering (SEC-MALS), which showed that Nup358 and BicD2 form a 2:2 complex (*Noell et al., 2018*). The one-site-binding model and the 2:2 stoichiometry are in line with Nup358-min binding as a dimer to a single binding site on a BicD2 coiled-coil dimer, although we cannot exclude the possibility that two Nup358 monomers bind to two binding sites on BicD2, where both sites have the same dissociation constant $K_D$. We also attempted to fit models assuming multiple binding sites and $K_{Ds}$, but those are not supported by the ITC thermogram. The equilibrium dissociation constant $K_D$ was determined to be 1.7 ± 0.9 μM, in a similar range as observed for other BicD2/cargo complexes as well as to the previously published affinity of 0.4 μM, obtained for BicD2-CTD towards a larger fragment of Nup358 (residues 2006–2443, with the GST-tag intact) (*Noell et al., 2018*). An ITC titration of Nup358-min-GST (i.e., with the GST-tag intact, whereas the GST was cleaved off in the first experiment) with BicD2-CTD yielded a very similar affinity of 1.6 ± 1.0 μM (*Figure 1—figure supplement 1*), demonstrating that the GST-tag does not affect the binding affinity. These ITC results confirm the mapped boundaries of the minimal binding site. The ITC analysis finally revealed that the Nup358-BicD2 interaction is driven by a favorable enthalpy change (ΔH = –19.2 ± 0.7 kcal/mol), which overcomes the unfavorable entropy change (ΔS = –38.2 ± 5.8 cal/mol/K).

### The DD-BicD2-Nup358^min complex moves processively on microtubules

Single-molecule reconstitutions were used to determine if the adaptor-binding protein Nup358^min can bind and relieve BicD2 autoinhibition, which in turn allows dynein and dynactin (DD) to be recruited and activated for processive motion. Nup358^min (N) and full-length BicD2 (B) were labeled with two different color quantum dots (Qdots), and tissue-purified DD was unlabeled (*Figure 2a*). When dynein, dynactin, BicD2, and Nup358^min were all present (DDBN^min), 23% dual-color complexes were observed (*Figure 2—figure supplement 1a and c*). To increase complex formation, Nup358^min was artificially dimerized with a leucine zipper (Nup358^min-zip), which increased the number of dual-colored complexes to 35% (*Figure 2—figure supplement 1a and b*). The rationale for this strategy was based on our previous observation that dimerization of the *Drosophila* adaptor-binding protein Egalitarian enhanced its affinity for BicD and bypassed the requirement for mRNA cargo for BicD activation

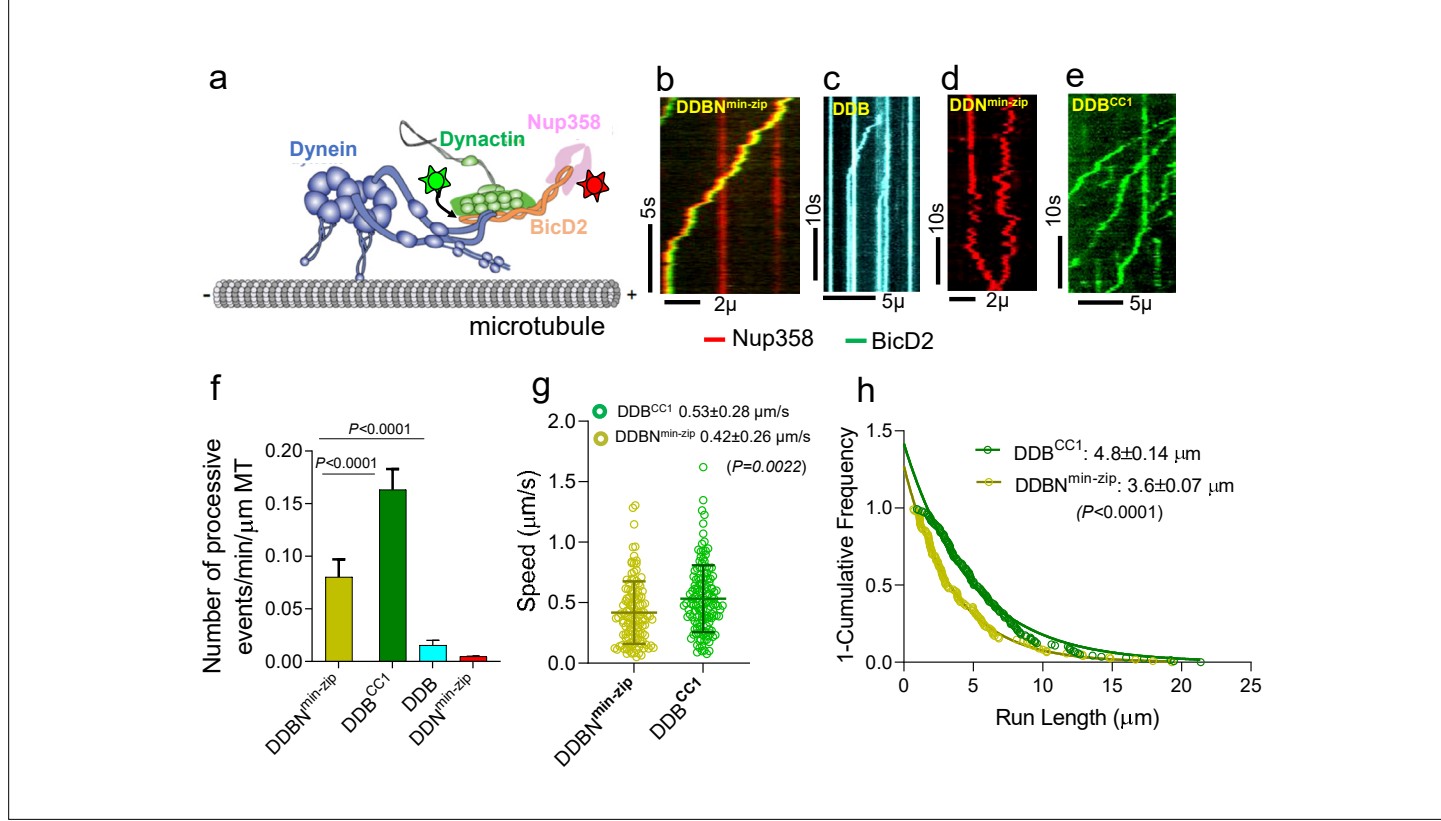

**Figure 2.** Nup358^min-zip is capable of forming a dynein-dynactin-BicD2-Nup358^min-zip complex (DDBN^min-zip) that is activated for processive motility. (**a**) Schematic of a DDBN^min-zip complex bound to a microtubule. BicD2 and Nup358 are shown labeled with two different color quantum dots (Qdot; stars). (**b**) Representative kymograph of the DDBN^min-zip complex moving processively on microtubules. Nup358^min-zip was labeled with a 655 nm Qdot and BicD2 with a 525 nm Qdot. (**c**) Dynein-dynactin-BicD2 (DDB) complexes bind to microtubules but show very little processive motion. (**d**) Dynein-dynactin-Nup358^min-zip complexes (DDN^min-zip) showed diffusive movement on microtubules. Nup358^min-zip was labeled with a 655 nm Qdot. (**e**) As a control, processive motion of dynein-dynactin in the presence of the N-terminal domain of BicD2 (DDB^CC1), with BicD2^CC1 labeled with a 525 nm Qdot. (**f**) Bar graph showing the number of processive events of DDBN^min-zip per min per micrometer microtubule (MT) length. As controls, number of processive events are shown for active DDB^CC1, DDB, and DDN^min-zip (**g, h**) Speed and run length of DDBN^min-zip (yellow) were compared with the constitutively active complex DDB^CC1 (green). Speeds for the two complexes are significantly different (p<0.0022, unpaired *t*-test with Kolmogorov–Smirnov test), as are the run lengths (p<0.0001, Kolmogorov–Smirnov test). See also *Figure 2—figure supplement 1*. For each panel, data were obtained from three independent experiments and two protein preparations.

The online version of this article includes the following video, source data, and figure supplement(s) for figure 2:

**Source data 1.** The number of processive events of DDBN^min-zip compared with DDB^CC1, DDB, and DDN^min-zip.

**Source data 2.** Speed of DDBN^min-zip compared with the constitutively active complex DDB^CC1.

**Source data 3.** Run length of DDBN^min-zip compared with the constitutively active complex DDB^CC1.

**Figure supplement 1.** Dimerization of Nup358^min increases the formation of DDBN^min-zip complexes.

**Figure 2—video 1.** Dynein-dynactin-BicD2-Nup358^min-zip (DDBN^min-zip) is indicated by dual-color quantum dot (Qdot) (yellow) moving on microtubule tracks (not always seen due to photobleaching) at 2 mM MgATP.

https://elifesciences.org/articles/74714/figures#fig2video1

**Figure 2—video 2.** Fewer dynein-dynactin-BicD2 (DDB) complexes (green) show processive events compared to DDBN^min-zip (see *Figure 2f*).

https://elifesciences.org/articles/74714/figures#fig2video2

**Figure 2—video 3.** Dynein-dynactin-BicD2^CC1 (DDB^CC1) complex (green) moving on microtubule tracks at 2mM MgATP at 2mM MgATP.

https://elifesciences.org/articles/74714/figures#fig2video3

**Figure 2—video 4.** Source video (replicate of *Figure 2—video 1*).

https://elifesciences.org/articles/74714/figures#fig2video4

**Figure 2—video 5.** Source video (replicate of *Figure 2—video 1*).

https://elifesciences.org/articles/74714/figures#fig2video5

*Figure 2 continued on next page*

*Figure 2 continued*

**Figure 2—video 6.** Source video (replicate of *Figure 2—video 1*).
https://elifesciences.org/articles/74714/figures#fig2video6

**Figure 2—video 7.** Source video (replicate of *Figure 2—video 2*).
https://elifesciences.org/articles/74714/figures#fig2video7

**Figure 2—video 8.** Source video (replicate of *Figure 2—video 2*).
https://elifesciences.org/articles/74714/figures#fig2video8

**Figure 2—video 9.** Source video (replicate of *Figure 2—video 2*).
https://elifesciences.org/articles/74714/figures#fig2video9

**Figure 2—video 10.** Source video (replicate of *Figure 2—video 3*).
https://elifesciences.org/articles/74714/figures#fig2video10

**Figure 2—video 11.** Source video (replicate of *Figure 2—video 3*).
https://elifesciences.org/articles/74714/figures#fig2video11

**Figure 2—video 12.** Source video (replicate of *Figure 2—video 3*).
https://elifesciences.org/articles/74714/figures#fig2video12

**Figure 2—video 13.** Source video.
https://elifesciences.org/articles/74714/figures#fig2video13

**Figure 2—video 14.** Source video (replicate of *Figure 2—video 13*).
https://elifesciences.org/articles/74714/figures#fig2video14

**Figure 2—video 15.** Source video (replicate of *Figure 2—video 13*).
https://elifesciences.org/articles/74714/figures#fig2video15

(*Sladewski et al., 2018*). All further single-molecule reconstitutions thus used the dimerized version of Nup358-min (Nup358$^{min-zip}$) because of its enhanced affinity for BicD2.

The dynein-dynactin-BicD2-Nup358$^{min-zip}$ (DDBN$^{min-zip}$) complex exhibited robust processive motion on surface-immobilized microtubules (MTs), implying that Nup358$^{min-zip}$ relieves BicD2 autoinhibition to allow dynein activation (*Figure 2b*, *Figure 2—video 1*). As controls, DDB was shown to support a few processive events (*Figure 2c*, *Figure 2—video 2*), and DDN$^{min-zip}$ exhibited only diffusive motion (*Figure 2d*, *Figure 2—video 13*). Nup358$^{min-zip}$ alone does not bind to microtubules (*Figure 2—figure supplement 1d*), and thus Nup358$^{min-zip}$ shows diffusive MT binding by virtue of an interaction with DD. The number of processively moving DDBN complexes on MTs is ~5.3 times higher than that of DDB complexes (*Figure 2f*). The speed and run length of all dual-color DDBN$^{min-zip}$ complexes were analyzed, with the run length obtained from cumulative distribution analysis with a one-phase exponential decay fit, and speed determined from the mean ± standard deviation (SD). The speed and run length of DDBN$^{min-zip}$ were quantified as 0.42 ± 0.26 μm/s (n = 123) and 3.6 ± 0.07 μm (n = 123), respectively (*Figure 2g and h*). These motile properties of DDBN$^{min-zip}$ were compared with that of the DDB$^{CC1}$ complex (B$^{CC1}$, the N-terminal coiled-coil 1 domain of BicD2), a well-established fully active complex (*McKenney et al., 2014*; *Sladewski et al., 2018*; *Figure 2e and f*, *Figure 2—video 3*). The speed and run length of the DDB$^{CC1}$ complex were 0.53 ± 0.28 μm/s (n = 137) and 4.8 ± 0.14 μm (n = 137), respectively, which are significantly faster (p=0.0022) and longer (p<0.0001 than that of DDBN$^{min-zip}$; *Figure 2g and h*). The number of processive events of DDBN$^{min-zip}$ on MTs is approximately half that of the active DDB$^{CC1}$ complex (*Figure 2f*). Importantly, the directed motion of DDBN$^{min-zip}$ is very different from the autoinhibited DD complex, which shows only diffusive movement on MTs (*Sladewski et al., 2018*). The enhanced number of processive events only in the presence of both BicD2 and Nup358$^{min-zip}$ suggests that Nup358$^{min-zip}$ functionally relieves the autoinhibition of BicD2 so that DD can bind and move processively on MTs.

## NMR titration mapped the BicD2-binding site to the N-terminal half of Nup358-min

Because our single-molecule processivity assays confirmed that the Nup358-min domain forms a DDBN complex that is activated for processive motility, we characterized the BicD2-binding site on Nup358-min, employing solution NMR, which can provide atomic resolution information for protein interactions in the native solution state. First, backbone assignment of Nup358-min was carried out

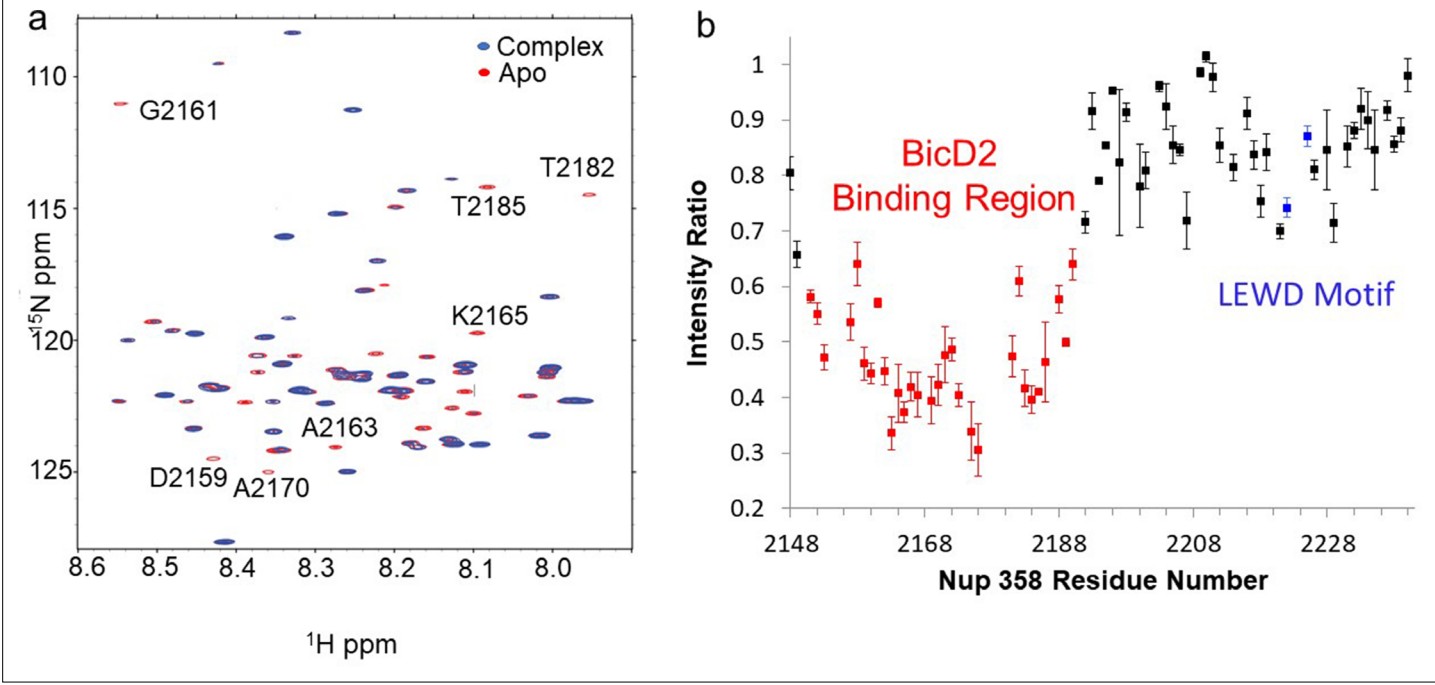

**Figure 3.** Nuclear magnetic resonance (NMR) titration mapped the BicD2-binding site to the N-terminal half of Nup358-min. NMR mapping of Nup358 regions involved in BicD2-CTD binding was performed by titration of $^{15}$N-labeled Nup358-min with BicD2-CTD. (**a**) The HSQC spectrum of a 1:1 Nup358-min/BicD2-CTD complex (blue) is overlaid on that of apo-Nup358-min (red). The full HSQC assignment is shown in **Figure 3—figure supplement 1**. Many peaks disappeared in the complex spectrum (selected peaks labeled by residue name and number), indicating that BicD2 binding causes slow to intermediate chemical exchange on the NMR time scale. (**b**) Plot of the peak intensity change vs. the residue number of Nup358. The peak intensities I, as determined by peak heights, of the 1:1 BicD2-CTD/Nup358-min complex spectrum were divided by the peak intensities of the apo-Nup358 spectrum ($I_0$) to obtain $I/I_0$. Data points with an $I/I_0$ of 0.65 or lower are colored red. This plot shows that the N-terminal half of Nup358-min is largely responsible for BicD2 binding. The peak intensities corresponding to the LEWD sequence motif (colored blue) in Nup358, which mediates binding of KLC2, are not affected by BicD2 binding. For the full titration results, please see **Figure 3—figure supplement 2**.

The online version of this article includes the following source data and figure supplement(s) for figure 3:

**Source data 1.** Source data for NMR titration plot.

**Figure supplement 1.** Fully assigned $^{15}$N-$^1$H HSQC nuclear magnetic resonance (NMR) spectrum for Nup358-min.

**Figure supplement 2.** Three points of Nup358:BicD2 titration.

**Figure supplement 3.** TALOS-N demonstrates random coil for Nup358-min.

using standard triple resonance experiments, 3D HNCO (**Kay et al., 1990**), HNCACO (**Clubb et al., 1992**), HNCA (**Kay et al., 1990**), HNCACB (**Grzesiek and Bax, 1992**), and CBCACONH (**Grzesiek and Bax, 2002**). 82 of the 89 total backbone amides were assigned in the $^1$H-$^{15}$N HSQC (**Figure 3—figure supplement 1**). Then, $^{15}$N-labeled Nup358-min was titrated with unlabeled BicD2-CTD. **Figure 3a** shows the overlay of the HSQC spectrum of apo Nup358-min with that of the complex, for which $^{15}$N-labeled Nup358-min and unlabeled BicD2-CTD were mixed at a 1:1 molar ratio. The HSQC spectrum of the apo Nup358-min is characterized by a lack of dispersion, consistent with an IDP (**Figure 1—figure supplement 1**, **Figure 3—figure supplement 1**). Analysis of backbone chemical shifts (CA, CB, CO, N, and HN) of Nup358-min by TALOS-N (**Shen and Bax, 2013**) conclusively demonstrates that Nup358-min is a random coil in the apo state (**Figure 3—figure supplement 3**). Although the addition of BicD2-CTD resulted in little peak movement in the HSQC, significant peak intensity changes were observed for the N-terminal half of Nup358-min (**Figure 3a and b**). This points to a relatively wide binding region undergoing intermediate to slow exchange on the NMR time scale due to ligand binding.

The Nup358-min domain also contains the previously published KLC2-binding LEWD motif (**Cui et al., 2019**), but the BicD2-binding site determined here is separated from the LEWD motif by over 30 residues (**Figure 3b**). Residues 2192–2240 of Nup358 show little change in NMR signal between the apo sample and the complex (**Figure 3b**), indicating that the LEWD motif is not involved in BicD2

binding. Therefore, kinesin-1 and the dynein adaptor BicD2 may bind to separate but spatially close binding sites on the cargo Nup358. The proximity of these motor recognition sites may serve to enable the interaction between dynein and kinesin machineries for the regulation of bidirectional motility and precise positioning of the nucleus.

## CEST suggests a coil-to-α-helix transition at the BicD2-Nup358 interface

Although our results from the HSQC titrations establish the general region of the BicD2 interface in Nup358, the chemical shifts of the bound state were not obtained due to peak disappearance in the HSQC upon BicD2 addition. To further characterize the BicD2/Nup358 interface with NMR, chemical exchange saturation transfer (CEST) experiments were performed. CEST has recently emerged as a powerful technique in solution NMR for measuring the chemical shifts of NMR invisible states (*Vallurupalli et al., 2012*; *Charlier et al., 2017*), such as the BicD2-CTD-bound state of Nup358-min, whose resonances are absent from the $^{15}$N-HSQC (*Figure 3a and b*). In CEST, when the invisible state is saturated by a weak and long radiofrequency (RF) pulse ($B_1$), the saturation of the bound state will be transferred to the free state due to Nup358-min dissociating from the Nup358-min-BicD2 complex, causing a dip in peak intensity when $B_1$ is on resonance with the bound state chemical shift, generating the minor dip in the CEST curve. If the chemical shifts of bound and unbound state are significantly different, which is, for example, the case for interface residues or residues that undergo structural changes upon complex formation, this difference gives rise to a double-dip appearance in the CEST profile (*Figure 4c and d*). The major dip in the profile will be observed at the chemical shift of the free or unbound state (when the $B_1$ matches the chemical shift of unbound resonance, the 'visible state'). The second, smaller dip in the profile peak will be observed at the chemical shift of the bound state (due to saturation transfer from the 'invisible state'). In contrast, residues that are not located at the complex interface or do not undergo structural changes upon complex formation will result in resonances without chemical exchange and the CEST profile will only show a single dip (*Figure 4a and b*), corresponding to the chemical shift of the unbound state.

We performed CEST experiments on both the $^{15}$N amide and the $^{13}$C carbonyl (C') resonances. In $^{15}$N CEST NMR experiments, the following residues were identified as interface residues or residues that undergo structural transition because of the double-dip appearance in the CEST profile: A2163, A2164, K2165, L2166, I2167, K2178, and L2184 (*Figure 4—figure supplement 1*). For these residues, CEST indicates a significant chemical shift difference between their bound and unbound states. Using the program RING NMR Dynamics (*Beckwith et al., 2020*), the CEST data were fit to obtain chemical shift difference between free and bound state ($\Delta\delta$), the exchange rate ($k_{ex}$), and the fractional minor population ($P_b$) (see *Table 1*, *Appendix 1—table 1a and b*). The weighted average of the $^{15}$N CEST fit parameters resulted in $k_{ex} = 200 \pm 70$ s$^{-1}$, consistent with slow to intermediate exchange ($k_{ex} < \Delta\omega$) on an NMR time scale. The weighted average of the minor population ($P_b$) is 0.04 ± 0.01, which is consistent with our sample preparation at a molar ratio of 1:20 for BicD2-CTD:$^{15}$N-Nup358-min. Many peaks in the $^{15}$N-CEST suffered from low signal-to-noise ratio and resonance overlap common in IDPs, preventing the determination of additional chemical shifts of invisible states. To overcome this problem, we further carried out $^{13}$C'-CEST experiments based on HNCO, which correlates amides (HN) with the carbonyl (CO) of the *preceding* residue. Here, the saturation pulse $B_1$ is on the carbonyl, and the intensity change due to saturation is still reported in an $^{15}$N-HSQC-like 2D spectrum. This method provided additional data for residues missed with the $^{15}$N CEST due to resonance overlap or low signal-to-noise ratio. For example, the F2183 peak in $^{15}$N-$^{1}$H HSQC was overlapped. However, the L2184 peak, a well-resolved signal with high S/N, showed a CEST effect from the saturation of the F2183 carbonyl in HNCO-based $^{13}$C'-CEST. With the $^{13}$C HNCO-CEST experiment, we were able to observe minor states in the following residues: R2162, A2164, K2165, L2166, R2169, E2171, E2172, L2177, K2181, F2183, and L2184 (*Figure 4—figure supplement 2*, *Appendix 1—table 1c*). Thus, with CEST, we were able to further map the binding region to residues 2162–2184 (*Figure 4e*). Fitting of HNCO-CEST profiles with RING Dynamics resulted in a global exchange rate of 270 ± 50 s$^{-1}$, in reasonable agreement with the value of $k_{ex}$ from $^{15}$N CEST. $k_{ex}$ obtained here is much slower in time scale than the rate of coil-to-α-helix transition in model peptides (*Wang et al., 2004*). This suggests that the $k_{ex}$ obtained by CEST include contributions from other events, such as binding and oligomerization, in addition to coil-to-α-helix transition, consistent with similar observations for other binding studies (*Charlier et al., 2017*). The weighted average of the minor population ($P_b$) is 0.06 ± 0.01,

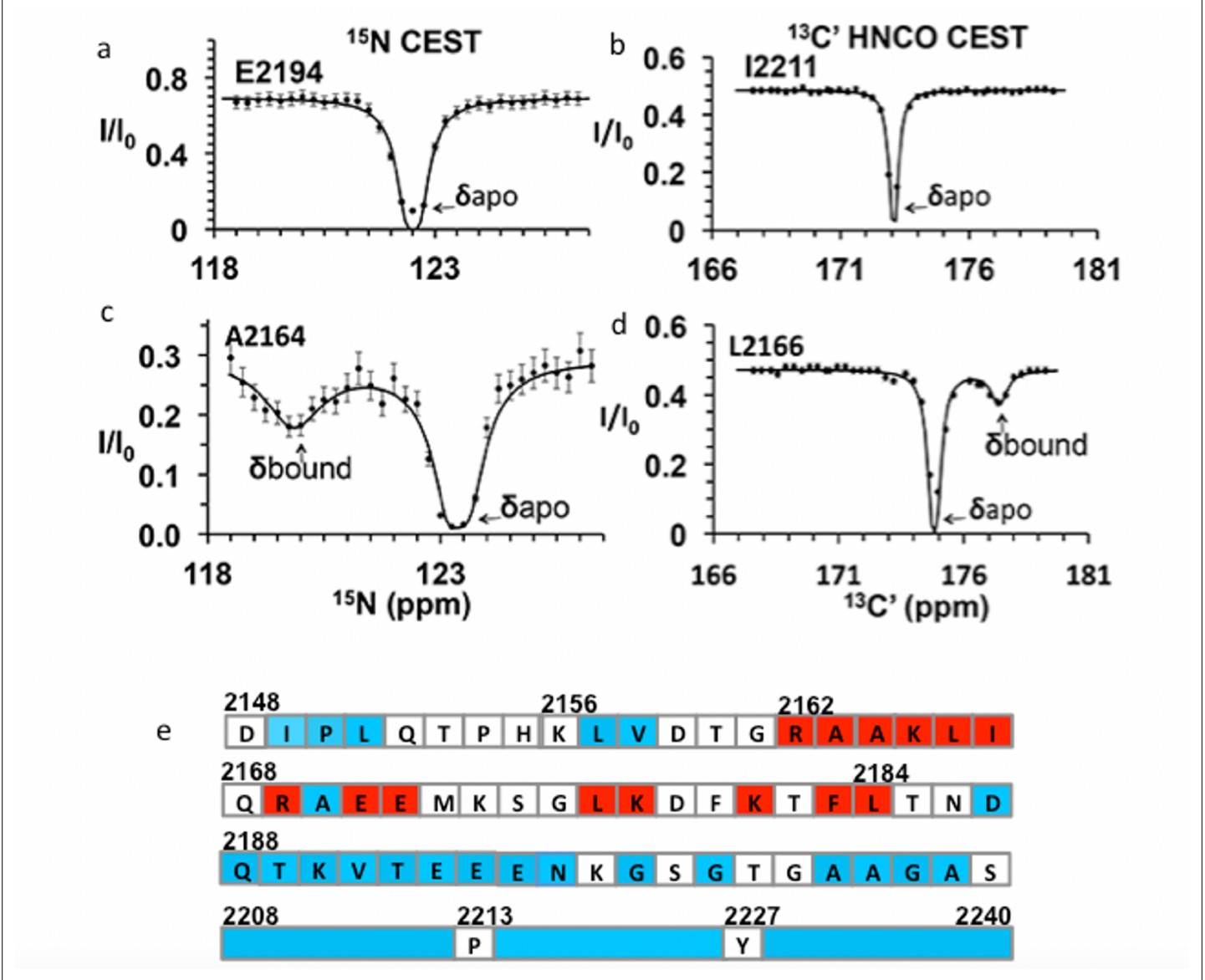

**Figure 4.** Chemical exchange saturation transfer (CEST) maps chemical shifts of nuclear magnetic resonance (NMR)-invisible, BicD2-bound state of Nup358. In the CEST profile curve, E2194 and I2211 have only a single dip in $^{15}$N-CEST (**a**) and $^{13}$C'-CEST (**b**), respectively, due to little chemical shift perturbation upon BicD2-CTD binding. This suggests that E2194 is not at the binding interface and does not undergo conformational transition upon BicD2-CTD binding. In contrast, A2164 shows not only a dip at the chemical shift of the apo HSQC assignment, but also a smaller dip at about 3 ppm to the left in $^{15}$N-CEST (**c**), indicating that A2164 is in the binding region and/or experiences a conformational transition upon complex formation. (**d**) L2166 has a major peak and then a minor peak about 2.5 ppm to the right. The magnitude and direction of the chemical shift difference suggest a transition from a random coil to an α-helix (see *Table 1*). (**e**) Summary of the CEST data obtained across the entire Nup358-min region. Red squares show residues with double dip in CEST profile. Their chemical shift changes from the apo to the bound state indicate that these red residues are part of the α-helical region at the Nup358/BicD2 interface. Blue-colored residues show a single dip in CEST profile, suggesting that these remain in random coil conformation. White represents the residues where data were not obtained due to low signal-to-noise ratio or resonance overlap. All the curves with double dips and a sampling of curves with only single dip fits are shown in *Figure 4—figure supplements 1–4*.

The online version of this article includes the following source data and figure supplement(s) for figure 4:

**Source data 1.** Source Data for CEST Curves.

**Figure supplement 1.** $^{15}$N chemical exchange saturation transfer (CEST) profiles for Nup358 residues showing exchange between random coil state and an α-helical state.

**Figure supplement 2.** $^{13}$C' chemical exchange saturation transfer (CEST) profiles for Nup358 residues showing exchange between random coil state and an α-helical state, showing the minor dip as well as the major dip, representing the chemical shift of the bound state.

*Figure 4 continued on next page*

*Figure 4 continued*

**Figure supplement 3.** [15]N chemical exchange saturation transfer (CEST) profiles for Nup358 residues showing NO exchange between the random coil state and an α-helical state.

**Figure supplement 4.** [13]C chemical exchange saturation transfer (CEST) profiles for Nup358 residues showing NO exchange between a random coil state and an α-helical state, showing only a single dip at the major chemical shift.

which is again consistent with the ratio in which we mixed the two proteins for the NMR sample. The 18 CEST profiles with double-dip appearance all resulted in chemical shift changes in the direction and magnitude as expected for a coil-to-α-helix transition (*Wishart, 2011*) (typically the chemical shift for [15]N becomes lower in the helical conformation, and higher for [13]C, as shown in *Table 1*). The CEST results suggest that residues 2162–2184 of Nup358 undergo coil-to-α-helix transition upon binding to BicD2, revealing that BicD2 recognizes Nup358 through a short 'cargo recognition α-helix,' which is embedded in an IDP domain.

To provide further evidence for the coil-to-α-helix transition, CD spectroscopy was carried out. The CD wavelength scans of the Nup358-min/BicD2-CTD complex had minima at 208 nm and 222 nm, characteristic for α-helical proteins (*Noell et al., 2019*; *Figure 5*). Such minima were absent in the CD spectra of Nup358-min as expected for an IDP. Notably, the minima at 208 and 222 nm were shifted towards more negative values in the complex compared to the sum of the individual spectra of Nup358-min and BicD2-CTD, suggesting that the α-helical content increases upon complex formation.

To quantify the increase of the α-helical content, we determined the difference between the molar ellipticity at 222 nm of the complex and of the sum of the individual proteins, $\Delta[\Theta]$, to be –4874 deg cm$^2$/dmol (*Figure 5*), which corresponds to a 14% ± 5% increase of the α-helical content based on a published thermal unfolding curve of the BicD2-CTD, which was used for calibration (*Noell et al., 2019*; *Figure 5*, *Figure 5—figure supplement 1*, and *Cui et al., 2020*). We recently determined the experimental error to be ~5% (corresponding to $\Delta[\Theta]$ of 2930 deg cm$^2$/dmol) (*Cui et al., 2020*); therefore, we estimate that the α-helical content increases by 14% ± 5% upon Nup358/BicD2 complex formation, confirming a structural transition from a coil to an α-helix.

## The minimal Nup358/BicD2 complex has a rod-like shape that is more compact than the individual proteins

Small-angle X-ray scattering (SAXS) experiments were carried out to obtain a low-resolution structure of the Nup358-min/BicD2-CTD complex. The quality of our SAXS data was confirmed by molar mass calculations: for Nup358-min, we determined a molar mass of 12.3 kDa (*Appendix 2—table 1*), which matches closely to the calculated mass of a monomer (10.6 kDa). For the Nup358-min/BicD2-CTD complex, we determined a molar mass of 47.6 kDa

**Table 1.** Chemical shift differences from chemical exchange saturation transfer (CEST) of Nup358/BicD2 and apo-Nup358 ($\Delta \delta$ $_{bound-apo}$) match closely to $\Delta \delta$ for coil-to-α-helix transition ($\Delta \delta$ $_{helix-coil}$)‡.

| Residue | $\Delta \delta$ $_{bound-apo}$ | $\Delta \delta$ $_{helix-coil}$ |
|---|---|---|
| A2163* | –3.6 | –2.2 |
| A2164* | –3.6 | –2.2 |
| K2165* | –0.8 | –1.3 |
| L2166* | –3.8 | –1.9 |
| I2167* | –5.7 | –1.2 |
| K2178* | –3.6 | –1.3 |
| L2184* | –5.5 | –1.9 |
| R2162† | 1.7 | 2.3 |
| A2164† | 2.8 | 1.7 |
| K2165† | 2.7 | 2.1 |
| L2166† | 2.6 | 1.6 |
| R2169† | 2.9 | 2.3 |
| E2171† | 2.2 | 2.2 |
| E2172† | 3.0 | 2.2 |
| L2177† | 0.9 | 1.6 |
| K2181† | 3.0 | 2.1 |
| F2183† | 3.3 | 1.5 |
| L2184† | 1.0 | 1.6 |

See also *Appendix 1—table 1*.

CD spectroscopy confirms formation of an α-helix in Nup358 upon binding to BicD2.

*Change in chemical shifts of amide [15]N.

†Change in chemical shifts of carbonyl [13]C.

‡Values taken from *Wishart, 2011*.

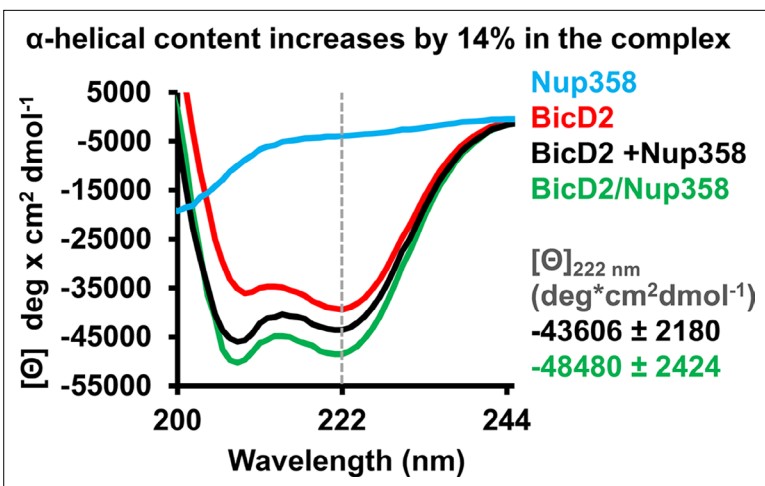

**Figure 5.** Circular dichroism (CD) spectroscopy confirms formation of an α-helix in the Nup358/BicD2 complex. CD wavelength scans of BicD2-CTD (red), Nup358-min (blue), and the Nup358-min/BicD2 complex (green) at 4°C are shown. The sum of the individual wavelength scans of Nup358-min and BicD2-CTD is shown in black. The mean residue molar ellipticity [Θ] versus the wavelength is shown. Experiments were repeated two or three times, and representative scans are shown. A characteristic feature of α-helical structures is a local minimum at 222 nm (dashed line). Based on the values for [Θ] at 222 nm (shown on the right) and based on our calibration curve (*Figure 5—figure supplement 1*), the Nup358/BicD2 complex has a 14% ± 5% increase in α-helical content compared to the sum of the spectra of BicD2 and Nup358. See *Figure 5—figure supplement 1* and *Figure 5— source data 1*.

The online version of this article includes the following source data and figure supplement(s) for figure 5:

**Source data 1.** CD spectra.

**Figure supplement 1.** Calibration curve for the circular dichroism (CD) signal, based on a published melting curve of BicD2-CTD.

---

(*Appendix 2—table 1*), which matches closely to the expected mass of a Nup358-min/BicD2-CTD complex with a 2:2 stoichiometry (calculated molar mass of 43.0 kDa). These findings are in line with our previously published SEC-MALS data, which suggest that apo Nup358-min forms monomers, while the Nup358/BicD2 complex has a 2:2 stoichiometry (*Cui et al., 2019*; *Noell et al., 2018*).

A Kratky plot of the SAXS profiles further confirmed that Nup358-min is intrinsically disordered as the signal increases at high q values instead of approaching zero (*Figure 6a*). In contrast, the Kratky plots of the BicD2-CTD and the Nup358-min/BicD2-CTD are bell-shaped, suggesting that they are folded (*Figure 6a*). Thus, Nup358-min becomes more compact upon complex formation with BicD2-CTD, consistent with the coil-to-α-helix transition observed from NMR and CD spectroscopy.

The pair distance distribution function p(r) derived from the SAXS profile of the Nup358-min/BicD2-CTD complex has a peak that decays with a linear slope, which is characteristic of elongated, rod-like structures (*Figure 6b*). Since the width of a rod remains similar throughout its length, the sum of all pair distances will have a linear slope. The maximum particle diameter $D_{Max}$ of the complex is 190 Å, which is identical to the $D_{Max}$ that was determined for the BicD2-CTD sample (*Appendix 2— table 1*), suggesting that the overall length of the rod structure does not change. Bead models of the Nup358-min/BicD2 complex were reconstructed from the p(r) functions and confirm that the complex has a flexible, rod-like structure (*Figure 6c*). The normalized spatial discrepancy (NSD) is 0.7, suggesting the structural convergence and homogeneity of bead models reconstructed from SAXS profiles (*Appendix 2—table 2*).

Taken together, these SAXS data suggest that the complex has a 2:2 stoichiometry and is more compact than the apo state, further validating the coil-to-α-helix transition in the Nup358/BicD2 complex.

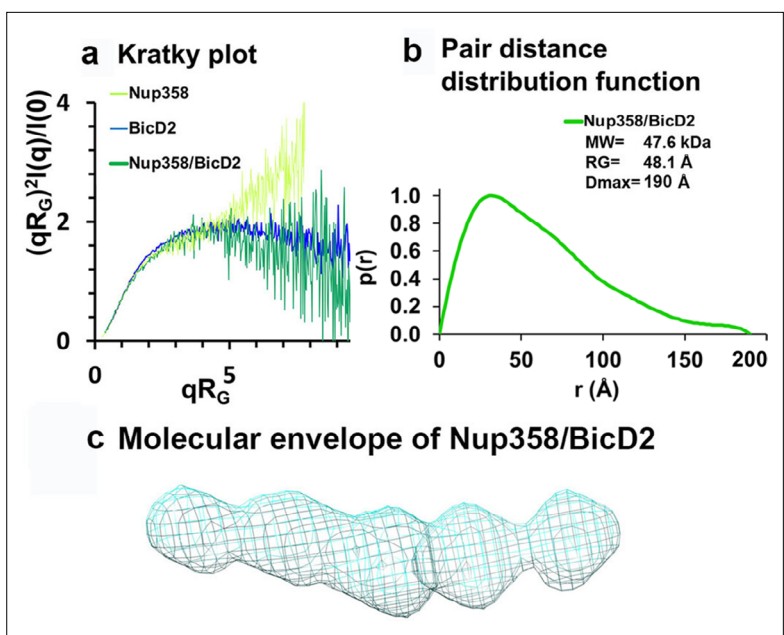

**Figure 6.** Low-resolution structures determined by small-angle X-ray scattering (SAXS) confirm that the complex has a rod-like shape that is more compact than the individual proteins. (**a**) Dimensionless Kratky plots of the SAXS data collected from the minimal Nup358/BicD2 complex, from Nup358-min and BicD2-CTD (q: scattering vector; $R_G$: radius of gyration; I(q): scattering intensity). (**b**) The pair distance distribution function p(r) of the Nup358-min/BicD2-CTD complex was derived from the scattering intensity profile (*Figure 6—figure supplement 1*). The molecular weight MW (*Rambo and Tainer, 2013*), the radius of gyration $R_G$ from the Guinier plot, and the largest dimension of the particle $D_{max}$ are shown. (**c**) Refined bead model 3D reconstruction of the Nup358-min/BicD2-CTD complex (cyan mesh). The statistical analysis and supporting SAXS data are summarized in *Figure 6—figure supplement 1–4* and Appendix 2. See also *Figure 6—source data 1*.

The online version of this article includes the following source data and figure supplement(s) for figure 6:

**Source data 1.** Small-angle X-ray scattering (SAXS) data.

**Figure supplement 1.** SAXS plots for Nup358-min/BicD2-CTD complex.

**Figure supplement 2.** SAXS plots for Nup358-min.

**Figure supplement 3.** SAXS plots for BicD2-CTD.

**Figure supplement 4.** The pair distance distribution p(r) functions of the Nup358-min/BicD2-CTD complex, Nup358-min, and BicD2-CTD.

## Nup358 mutagenesis validates α-helix formation at the Nup358/BicD2 interface

Our CEST, CD, and SAXS data provided strong evidence that residues 2162–2184 of Nup358 undergo a structural transition from random coil-to-α-helix upon complex formation with BicD2.

To confirm our results, we also carried out alanine mutagenesis for all residues of the Nup358 α-helix and assessed binding of the mutated GST-tagged Nup358-min to BicD2-CTD by pull-down assays (*Figure 7*, *Figure 7—figure supplement 1*). Eleven mutants displayed diminished binding, which are interspaced throughout the α-helix. An α-helical wheel representation was created, which revealed that these interface residues are clustered on one side of the α-helical wheel (*Figure 7c*, red). The mutagenesis experiments are consistent with our results from NMR spectroscopy. While interface residues and residues that undergo the transition from coil-to-α-helix will show a chemical shift change, not all residues in the newly formed α-helix in Nup358-min will be at the binding interface; therefore, the mutagenesis data provides additional information. Notably, all interface residues that were identified from mutagenesis (*Figure 7d*, red) also had a double dip in the CEST profile (*Figure 7d*, red) or were not assessed (*Figure 7d*, white). Furthermore, removal of residues 2163–2166 (AAKL) of Nup358-min (ΔAAKL), which had double dips in CEST curves, virtually abolishes the interaction with the BicD2-CTD (*Figure 7—figure supplement 1b*). We also removed the α-helix from

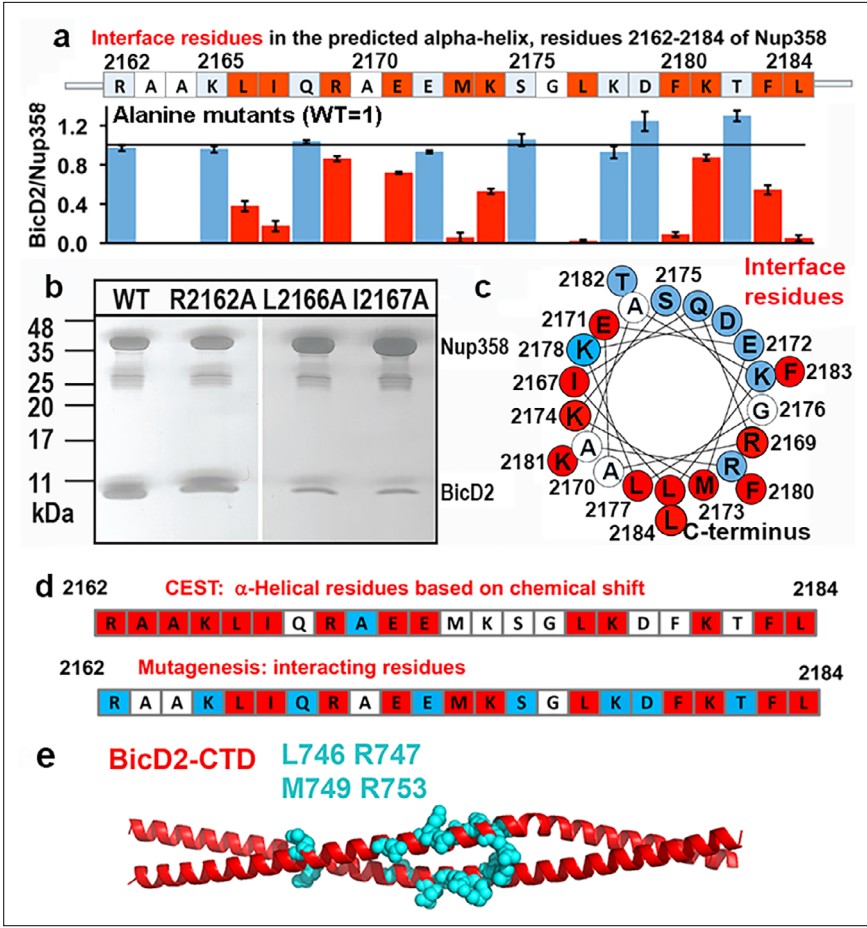

**Figure 7.** Mutagenesis of the Nup358 cargo recognition helix. All residues of the Nup358 α-helix were mutated to alanine, and binding was assessed by GST-pull-down assays with purified Nup358-min-GST followed and BicD2-CTD. The elution fractions were analyzed by SDS-PAGE, and the intensities of the gel bands were quantified to obtain the ratio of bound BicD2-CTD/Nup358-min, normalized respective to the wild-type (*Figure 7—figure supplement 1*). The binding ratio was averaged from three independent experiments, and the error was calculated as the standard deviation. Eleven interface residues were identified, which are colored red and show a reduction of the binding ratio upon mutation by at least three times the standard deviation. In addition, interface residues were required to have a binding ratio that was reduced by 13% or more compared to the wild-type. Alanine and glycine residues were not assessed and are colored white. Residues for which mutations do not affect binding are colored light blue. (**a**) The sequence of the predicted Nup358 α–helix is shown above a bar graph of the ratios of bound BicD2 to Nup358 from the alanine mutant pull-down assays (WT = 1, indicated by the horizontal black line). (**b**) Representative SDS-PAGEs of elution fractions of the pull-down assays. A representative full dataset is shown in *Figure 7—figure supplement 1*. Note that a small gel band at 25 kDa represents GST. (**c**) Helical wheel representation for (**a**). (**d**) Comparison of helical residues in Nup358-BicD2 complex identified by chemical exchange saturation transfer (CEST) and results from pull-down assay of mutants. Red: CEST-positive (double dip in CEST profile) or strong reduction in binding from the pull-down assay with alanine mutation at this residue; blue: CEST-negative (single dip in CEST profile) or no effect in mutagenesis; white: data not available. (**e**) The structure of the *Hs* BicD2-CTD (*Noell et al., 2019*) is shown in red cartoon representation. Four Nup358/BicD2 interface residues that were previously identified by mutagenesis are shown in cyan spheres representation (*Terawaki et al., 2015*). See also *Figure 7—figure supplement 1* and *Figure 7—source data 1–3*.

The online version of this article includes the following source data and figure supplement(s) for figure 7:

**Source data 1.** Quantification of the pull-down assay.

**Source data 2.** SDS-PAGE shown in *Figure 7*.

**Source data 3.** SDS-PAGEs shown in *Figure 7—figure supplement 1* and SDS-PAGEs that were used for the quantification shown in *Figure 7*.

**Figure supplement 1.** Interface residues of BicD2 and Nup358 were identified by mutagenesis.

**Figure supplement 1—source data 1.** BicD2-CTD calibration curve for quantification.

Nup358-min, and as expected, the resulting Nup358-fragment (residues 2185–2240) shows virtually no interaction with the BicD2-CTD (*Figure 7—figure supplement 1a*, last lane). Finally, we successfully assembled a minimal complex of the BicD2-CTD with a Nup358 fragment (residues 2158–2199) that only contained the BicD2-binding site deduced from NMR, which confirmed our mapped binding site (*Figure 7—figure supplement 1c*). Furthermore, known BicD2 residues that are important for Nup358 interaction include L746, R747, M749, and R753 (*Terawaki et al., 2015*), suggesting a potential binding site for the Nup358 α-helix in the center of the BicD2-CTD coiled coil (*Figure 7e*).

## Nup358-BicD2 interface is essential for dynein activation

Five mutants that showed a reduced Nup358-min/BicD2-CTD interaction in the pull-down assays were selected for further characterization. Single-molecule-binding and processivity assays were used to further explore the impact of Nup358 mutants on dynein recruitment and activation of dynein motility. We assessed the interaction between BicD2 and the following Nup358$^{min-zip}$ mutants: I2167A, M2173A, L2177A, F2180A, and L2184A, in the context of the BN$^{min-zip}$ and the DDBN$^{min-zip}$ complex. The following parameters were quantified for WT and mutant Nup358$^{min-zip}$ constructs: (a) the number of single-molecule processive events of DDBN$^{min-zip}$ on MTs, (b) the percent complex formation with BicD2 alone or with the DDB complex, and (c) the speed and run length of DDBN$^{min-zip}$ on MTs.

The number of processively moving DDBN$^{min-zip}$ complexes on MTs was quantified as the number of dual-color Qdots moving per min per micrometer MT length (*Figure 8a*). The processive events of DDBN$^{min-zip}$ formed with the Nup358$^{min-zip}$ mutants I2167A, M2173A, F2180A, and L2184A Nup358$^{min-zip}$ were significantly lower than that formed with WT-Nup358$^{min-zip}$ (*Figure 8a*). DDBN$^{min-zip}$ formed with the Nup358$^{min-zip}$-L2177A mutant did not result in a lower number of processively moving complexes, suggesting that there is some plasticity at the Nup358/BicD2 interface.

We next assessed how these Nup358$^{min-zip}$ mutants affected the BicD2-Nup358$^{min-zip}$ (BN) interaction in the absence or presence of DD. BicD2 and WT or mutant Nup358$^{min-zip}$ were labeled with different color Qdots, and the number of dual-color BN$^{min-zip}$ and DDBN$^{min-zip}$ complexes nonspecifically bound to a coverslip was quantified using the total internal reflection fluorescence (TIRF) microscopy (*Figure 8—figure supplement 1*). Nup358$^{min-zip}$ binds BicD2 alone moderately well, and the percentage of BN was reduced further for the Nup358$^{min-zip}$ mutants (*Figure 8b*, red). We also quantified how the presence of DD influences the BicD2-Nup358$^{min-zip}$ interaction. Strikingly, in the presence of DD, the formation of DDBN$^{min-zip}$ complexes was about two times higher than the BN complexes (*Figure 8b*, gray). The mutant Nup358$^{min-zip}$ again showed reduced complex formation relative to WT (*Figure 8b*, gray). The difference between DDBN$^{min-zip}$ and BN$^{min-zip}$ indicates that DD enhances the BicD2-Nup358 interaction.

The motion of DDBN$^{min-zip}$ complexes formed with the five Nup358$^{min-zip}$ mutants was quantified and compared with DDBN$^{min-zip}$ containing WT-Nup358$^{min-zip}$. DDBN$^{min-zip}$ complexes with Nup358$^{min-zip}$ mutants showed slower speed and shorter run length compared with WT-DDBN$^{min-zip}$ (*Figure 8c and d*). There were too few dual-colored complexes formed with the I2167A mutant to allow speed and run length quantification. Compromised motility in the presence of the five characterized Nup358$^{min-zip}$ mutants suggests the formation of a less stable DDBN complex, even for the L2177A mutant that showed a similar number of processive runs as WT Nup358$^{min-zip}$. The functional importance of the Nup358 cargo recognition-α-helix is validated by the impaired motility observed with point mutants of this α-helix.

## Discussion

Here, we elucidate the molecular details of the interface between the dynein activating adaptor BicD2 and the nuclear pore protein Nup358, which links BicD2 to the cell nucleus. First, we reconstituted a minimal Nup358-min/BicD2-CTD complex with µM affinity. Single-molecule processivity assays revealed that a dimerized Nup358-min domain forms a complex with dynein/dynactin/BicD2 (DDBN) that shows robust processive motility on microtubules. Similarly, the small GTPase Rab6a$^{GTP}$ that links BicD2 to membranous cargo promotes processive motility (*Huynh and Vale, 2017*), suggesting that both adaptor-binding proteins act to release the autoinhibition of BicD2. Dimerization of Nup358-min enhances interaction with BicD2 and with the DDB complex, suggesting that activation is linked to oligomerization, which for Rab6$^{GTP}$ could occur by clustering on the vesicular membrane.

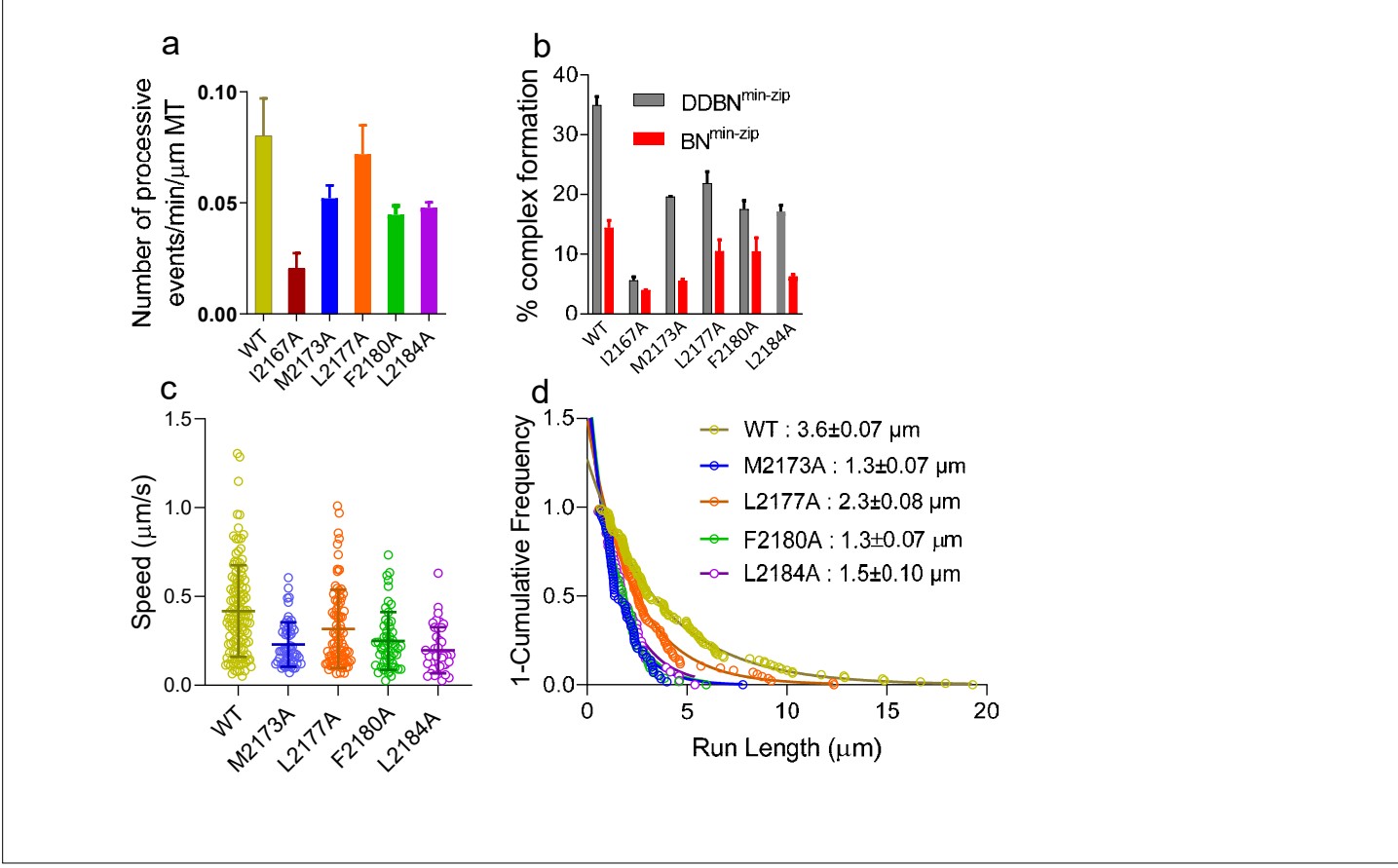

**Figure 8.** Nup358 point mutations that diminish the interaction with BicD2-CTD also diminish the formation of the DDBN complex. (**a**) Bar graph of the processive events of DDBN$^{min-zip}$ complexes per min per micrometer MT length. The number of processive events of DDBN$^{min-zip}$ complexes formed with WT-Nup358$^{min-zip}$ is significantly higher (p<0.0001) than those formed with the Nup358$^{min-zip}$ mutants I2167A, M2173A, F2180A, and L2184A but the same as with mutant L2177A (orange, p=0.09). The number of processive events of DDBN$^{min-zip}$ formed with WT-Nup358$^{min-zip}$ was 0.080 ± 0.0017/min/μm MT (n = 3, with n being the number of experiments). In contrast, these values for DDBN$^{min-zip}$ formed with the Nup358$^{min-zip}$ mutants I2167A, M2173A, L2177A, F2180A, and L2184A were 0.020 ± 0.007/min/μm (n = 2), 0.052 ± 0.006/min/μm (n = 2), 0.072 ± 0.013min/μm (n = 3), 0.044 ± 0.005/min/μm (n = 2), 0.048 ± 0.0023/min/μm (n = 2) respectively. (**b**) The presence of dynein-dynactin (gray) increases the formation of BicD2-Nup358$^{min-zip}$ complexes (red). The percent formation of BN$^{min-zip}$ was 14.41% (n = 580) for WT, 3.96% (n = 1142) for I2167A, 5.54% (n = 1877) for M2173A, 10.49% (n = 948) for L2177A, 10.47% (n = 412) for F2180A, and 6.31% (n = 1032) for L2184A Nup358$^{min-zip}$. N is the total number of quantum dots (Qdots) counted. In contrast, these values for DDBN$^{min-zip}$ complexes were 34.92% (n = 945), 5.62% (n = 986), 19.61% (n = 347), 21.87% (n = 778), 17.52% (n = 890), and 17.13% (n = 372), respectively. (**c**) Comparison of the speeds of DDBN complexes formed with WT-Nup358$^{min-zip}$ and mutant Nup358$^{min-zip}$ constructs. The speed of DDBN formed with Nup358$^{min-zip}$ mutants M2173A (0.23 ± 0.13 μm/s, n = 58; blue), L2177A (0.32 ± 0.22 μm/s; n = 86; orange), F2180A (0.25 ± 0.16 μm/s, n = 54; green), and L2184A (0.20 ± 0.13 μm/s, n = 41; purple) are significantly slower than that of DDBN-WT (p<0.0001 for mutants M2173A, F2180A, L2184A and p=0.0058 for L2177A), one-way ANOVA followed by Tukey's test. (**d**). The run length of DDBN$^{min-zip}$ formed with Nup358$^{min-zip}$-M2173A (1.3 ± 0.07 μm, n = 58), L2177A (2.3 ± 0.08 μm, n = 86), F2180A (1.3 ± 0.07 μm, n = 54), and L2184A (1.5 ± 0.10 μm, n = 41) are significantly shorter than that formed with WT-Nup358$^{min-zip}$ (3.6 ± 0.07 μm; n = 123; data taken from 2 hr) (with p<0.0001 for mutants M2173A, F2180A, L2184A and p=0.0036 for L2177A; one-way ANOVA followed by Tukey's test). See also *Figure 8—figure supplement 1*.

The online version of this article includes the following source data and figure supplement(s) for figure 8:

**Source data 1.** The number of processive events of DDBN$^{min-zip}$ complexes formed with WT-NUP358$^{min-zip}$ compared with DDBN$^{min-zip}$ formed with NUP 358$^{min-zip}$ mutants I2167A, M2173A, F2180A, and L2184A.

**Source data 2.** The presence of dynein-dynactin increases the formation of BicD2-NUP358$^{min-zip}$ complexes.

**Source data 3.** Comparison of the speeds of DDBN complexes formed with WT-NUP358$^{min-zip}$ and mutant NUP358$^{min-zip}$ constructs.

**Source data 4.** Comparison of the run lengths of DDBN complexes formed with WT-NUP358$^{min-zip}$ and mutant NUP358$^{min-zip}$.

**Figure supplement 1.** Nup358 point mutations that diminish the interaction with BicD2-CTD also diminish the formation of the DDBN$^{min-zip}$ complex.

NMR titration of BicD2-CTD into $^{15}$N labeled Nup358-min mapped the binding region to the N-terminal half of Nup358-min. Furthermore, we obtained chemical shifts for the CA and CB atoms for the apo state of Nup358-min, which confirm that the apo state is intrinsically disordered. Due to slow chemical exchange on the NMR time scale and fast relaxation in the BicD2-bound state of Nup358-min, this state cannot be directly observed in standard NMR experiments, and thus a powerful solution NMR technique termed CEST was applied to map the chemical shifts of the 'invisible state.' This approach not only identified key residues at the Nup358/BicD2 interface, but also showed that chemical shift changes upon binding are consistent with a coil-to-α-helix transition in Nup358-min. Notably, the coil-to-α-helix conformational change is supported by our results from CD spectroscopy and SAXS, and the BicD2/Nup358 interface was validated by mutagenesis. The observed time scale is also in line with other proteins undergoing coil-to-α-helix transitions (*Charlier et al., 2017*). However, it should be noted that in order to obtain higher-resolution structural information, the chemical shifts for the CA and CB atoms of Nup358-min in the bound state remain

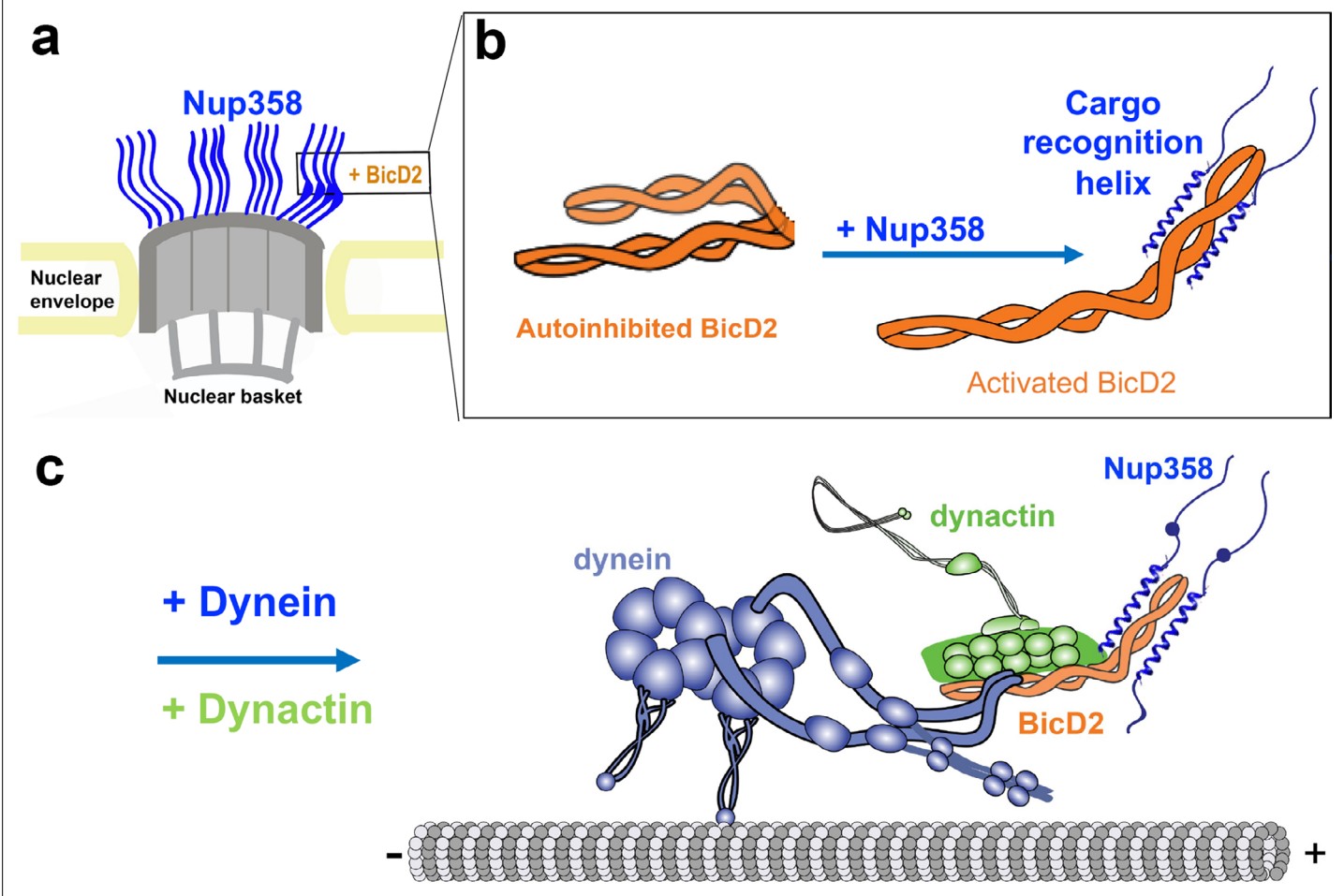

**Figure 9.** We propose that BicD2 recognizes its cargo through a short 'cargo recognition α-helix,' which may also be a structural feature that stabilizes the activated state of BicD2 for the recruitment of dynein and dynactin. (**a**) Cutaway view of half of an NPC. Each of the eight spokes of the NPC contains four molecules of Nup358 on the cytoplasmic side (i.e., 32 in total), which provide binding sites for dynein (via BicD2) and kinesin-1 (via KLC2). Nup358 makes up most of the mass of the cytoplasmic filaments of the NPC (*von Appen et al., 2015*; *Ori et al., 2013*). (**b**) Left panel: schematic representation of the looped, autoinhibited conformation of BicD2 that is formed in the absence of cargoes such as Nup358. Apo-Nup358 contains many intrinsically disordered regions (IDR), including the BicD2-binding region. Upon binding of Nup358 to BicD2, a short α-helix is formed at the Nup358/BicD2 interface, which is a structural feature that stabilizes BicD2 in the active state. Binding of Nup358 to BicD2 likely promotes loop opening, which activates BicD2 for dynein/dynactin recruitment (*Sladewski et al., 2018*). (**c**) Schematic representation of the proposed dynein/dynactin/BicD2/Nup358 complex (DDBN) bound to microtubules for processive motility for nuclear positioning. Nup358 has a LEWD motif that recruits kinesin-1 via KLC2. The binding sites for the opposite polarity motor kinesin-1 on Nup358 are indicated by the small blue sphere. The location of the LEWD motif that recruits kinesin-1 via KLC2 is indicated. We propose that BicD2 and KLC2 interact simultaneously with Nup358 for precise control of nuclear positioning.

to be determined in the future. Notably, our functional assays highlight the important role of the interaction between Nup358 and BicD2 in dynein activation. Single-molecule-binding assays showed that mutagenesis of the Nup358 cargo recognition α-helix diminished the interaction of Nup358-min with full-length BicD2 and decreased the formation of dynein/dynactin/BicD2/Nup358 complexes (DDBN). Furthermore, the mutant DDBN complexes that were formed exhibited reduced run length and speed reflecting formation of a less stable complex. Our data highlight the important role of dynein activating-adaptor/adaptor-binding protein interactions in activating and fine-tuning dynein motility.

Our data establish that BicD2 recognizes its cargo-binding protein Nup358 through an α-helix of ~28 residues. Coil-to-α-helix transitions and folding upon binding, as shown here, have been observed in many studies of IDP/IDR interactions and recognition (*Celestino et al., 2019*; *Charlier et al., 2017*). We propose that the dynein activating-adaptor BicD2 recognizes its cargoes through a 'cargo recognition α-helix' (*Figure 9*). It is conceivable that the cargo adaptors Rab6$^{GTP}$ and nesprin-2G are also recognized by similar short α-helices, although this needs to be confirmed by future experiments. Rab6$^{GTP}$ binds to BicD2 with 10-fold higher affinity compared to Rab6$^{GDP}$ (*Matanis et al., 2002*; *Bergbrede et al., 2009*; *Garcia-Saez et al., 2006*) and associated structural changes are found in the Switch I and Switch II regions of Rab6 (*Garcia-Saez et al., 2006*), which include short α-helices and intrinsically disordered domains and are possibly candidates for the BicD2-binding site. The mapped nesprin-2G-binding domain for BicD2 is also predicted to be largely intrinsically disordered (*Zhu et al., 2017*). Our data thus suggest a structural basis for cargo recognition by BicD2 and will enable further studies aimed at identifying cell cycle-specific regulatory mechanisms for distinct transport pathways that are facilitated by BicD2 (*Splinter et al., 2010*; *Gonçalves et al., 2020*; *Matanis et al., 2002*). For example, our Nup358 mutants that selectively target the interaction with BicD2 can be used to dissect biological roles of the Nup358/BicD2, Rab6/BicD2, and nesprin 2G/BicD2 pathways in distinct stages of brain development.

A key step in activating dynein motility is overriding the autoinhibited conformation of BicD2 to allow dynein recruitment. We and others have recently proposed that cargo binding activates BicD2 for dynein recruitment by inducing a coiled-coil registry shift in BicD2 (i.e., a vertical displacement of the two α-helices against each other by one helical turn) (*Liu et al., 2013*; *Terawaki et al., 2015*; *Noell et al., 2019*; *Cui et al., 2020*). In the absence of cargo, BicD2 forms an autoinhibited state that cannot recruit dynein, with the CTD masking the N-terminal dynein/dynactin-binding site (NTD). When cargo is loaded, BicD2 can directly link dynein with its activator dynactin, which is required to activate dynein for processive motility (*Splinter et al., 2012*; *Schlager et al., 2014b*; *Schlager et al., 2014a*; *Hoogenraad et al., 2003*; *Hoogenraad et al., 2001*; *McKenney et al., 2014*; *Urnavicius et al., 2018*; *Urnavicius et al., 2015*; *Sladewski et al., 2018*; *McClintock et al., 2018*; *Liu et al., 2013*; *Cui et al., 2020*). We recently showed that an F684I mutant of the *Drosophila (Dm)* homolog BicD can activate dynein/dynactin for processive motility in the absence of cargo. X-ray structures of the WT and F684I mutant as well as molecular dynamics simulations suggest that this activating mutation causes a coiled-coil registry shift in the *Dm* BicD-CTD (*Noell et al., 2019*; *Cui et al., 2020*). It is tempting to hypothesize that the cargo recognition helix of Nup358 intercalates into the coiled-coil of the BicD2-CTD to stabilize a coiled-coil registry shift in the Nup358/BicD2 complex, resulting in activation of BicD2 for dynein recruitment. This hypothesis remains to be experimentally confirmed.

Nup358-min binds more strongly to the dynein/dynactin/BicD2 complex compared to full-length BicD2 alone (*Figure 8b*). This observation supports the proposed loop-opening activation mechanism of BicD2 (*Hoogenraad et al., 2001*; *Sladewski et al., 2018*; *McClintock et al., 2018*; *Liu et al., 2013*; *Cui et al., 2020*; *Figure 9b and c*). In the full-length dynein/dynactin/BicD2/Nup358 (DDBN) complex, once Nup358 binding displaces the autoinhibited loop, dynein and dynactin bind and prevent its reformation. This is not true in the full-length BicD2/Nup358 complex, resulting in a lower apparent affinity than in the full complex. It is also possible that additional weak interfaces further stabilize the DDBN complex compared to Nup358/BicD2 or the DDB$^{CC1}$ complex. Here, we observed a direct interaction of Nup358 with DD that could contribute to enhanced affinity. It has been shown for several cargo adaptors that they interact weakly with the dynein light chains (*Tai et al., 1999*) and the dynein LIC1 forms a small interface with the so-called CC1 box of BicD2, which is required for the activation of processive motility (*Celestino et al., 2019*; *Lee et al., 2020*; *Sweeney and Holzbaur, 2018*). Cargoes and adaptors often interact through multiple interfaces with motors, and these additional interfaces in many cases enhance overall motility (*Fu and Holzbaur, 2014*; *Urnavicius et al., 2018*; *Urnavicius et al., 2015*; *Celestino et al., 2019*; *Lee et al., 2018*; *Sanger et al., 2017*; *Blasius et al., 2007*).

The affinity of the monomeric cargo Nup358-min for BicD2 is enhanced by dimerization by a leucine zipper, similar to what was previously observed for binding of the cargo Egalitarian for *Drosophila* BicD (*Sladewski et al., 2018*; *McClintock et al., 2018*). Thus, it is conceivable that a Nup358-min dimer binds to a BicD2 dimer, which is also in line with our ITC results, which reveal a single $K_D$. While Nup358-min is a monomer on its own, it oligomerizes to form 2:2 complexes with either the BicD2-CTD or the KLC2, and also forms a ternary complex with both BicD2-CTD and KLC2 with 2:2:2 stoichiometry (*Cui et al., 2019*; *Noell et al., 2018*). We thus propose that cargo adaptor-induced dimerization may potentially be a universal feature of the activation mechanism since BicD2 and KLC2 both form dimers in active dynein and kinesin-1 motors (*McKenney et al., 2014*; *Urnavicius et al., 2015*; *Cui et al., 2019*; *Cockburn et al., 2018*). Importantly, NMR titration showed that BicD2 binding has no effect on the NMR signal of the LEWD motif of Nup358 (*Figure 3*), which can interact directly with KLC2 (*Cui et al., 2019*). These data further suggest that the Nup358-min domain is potentially capable of recruiting both dynein and kinesin-1 machineries for bidirectional positioning of the nucleus, although this remains to be confirmed in the context of full-length proteins and intact motors. Intriguingly, our NMR data show that the BicD2-binding site is only separated by 30 residues from the LEWD motif that acts as KLC2-binding site. The proximity of these binding sites may play a role in coordinating motility for bidirectional transport (*Fu and Holzbaur, 2014*; *Hendricks et al., 2010*; *Wilson and Holzbaur, 2012*; *Splinter et al., 2010*; *Soppina et al., 2009*; *Encalada et al., 2011*; *Zhang et al., 2007*; *Cui et al., 2019*; *Belyy et al., 2016*; *Feng et al., 2020*; *Roberts et al., 2014*; *Gross et al., 2003*; *Derr et al., 2012*; *Ally et al., 2009*; *Kendrick et al., 2019*; *Furuta et al., 2013*). Another intriguing possibility is that if kinesin binding dimerizes Nup358, this may be the key initial step leading to BicD2 activation and recruitment of DD to form a bidirectional complex (*Figure 9*). The ternary Nup358/BicD2/KLC2 complex may be a model for other transport modules, in which opposite polarity motors such as dynein and kinesin-1 act together.

## Conclusion

Based on our data, we propose a structural basis for cargo recognition through the dynein adaptor BicD2. Our results establish that BicD2 recognizes its cargo Nup358 through a small 'cargo recognition α-helix' that is embedded in an IDR. This region undergoes a structural transition from a random coil to an α-helix upon binding to BicD2. In single-molecule TIRF assays, four single-point mutations within the cargo recognition helix significantly inhibited the interaction between BicD2 and Nup358$^{min-zip}$, and impaired run length and speed of the mutant DDBN$^{min-zip}$ complexes on microtubules. Our results may facilitate the identification of regulatory mechanisms for BicD2-dependent transport pathways, which are important for cell cycle control, brain and muscle development, and vesicle transport.

Activation of BicD2 is a key regulatory step for transport as it is required to activate dynein for processive motility. The cargo recognition α-helix may be a structural feature that stabilizes BicD2 in its activated state. We propose that binding of cargo induces a coiled-coil registry shift in BicD2, which promotes loop opening and activates BicD2 for dynein/dynactin binding. Notably, Nup358 interacts more strongly with the dynein/dynactin/BicD2 complex compared to BicD2 alone, which supports the loop-opening mechanism for activation. We also show that BicD2 and KLC2 bind to spatially close, but nonoverlapping binding sites on Nup358, supporting the hypothesis that Nup358 is capable of simultaneously recruiting dynein and kinesin-1 machineries to the nucleus for bidirectional transport.

# Materials and methods

## Key resources table

| Reagent type (species) or resource | Designation | Source or reference | Identifiers | Additional information |
|---|---|---|---|---|
| Strain, strain background (*Escherichia coli*) | Rosetta 2(DE3)-pLysS | Fisher Scientific | Cat# 714033 | |
| Strain, strain background (*E. coli*) | BL21-DE3-CodonPlus-RIL | Fisher scientific | Cat# 50-125-350 | |
| Recombinant DNA reagent | Human BicD2-CTD pet28a | GenScript Reference *Noell et al., 2019* doi:10.1021/acs.jpclett.9b01865 | Sequence encoding residues 715–804 of human BicD2 cloned into a pet28a vector via the NdeI and XhoI restriction sites | For the protein sequence expressed from this vector, see 'Supplementary methods' |

*Continued on next page*

*Continued*

| Reagent type (species) or resource | Designation | Source or reference | Identifiers | Additional information |
|---|---|---|---|---|
| Recombinant DNA reagent | Human Nup358-min pGEX-6P-1 | GenScript Reference *Noell et al., 2019* doi:10.1021/acs.jpclett.9b01865 | Sequence encoding residues 2148–2240 of human Nup358 cloned into a pGEX-6P-1 vector via the BamHI and XhoI restriction sites | For the protein sequence expressed from this vector, see 'Supplementary methods' |
| Recombinant DNA reagent | Nup358$^{-min}$-pGEX-6P-1 | GenScript This paper | Modified Nup358-min pGEX-6P-1 vector that includes a SNAP-tag at its C-terminal domain | For the protein sequence expressed from this vector, see 'Supplementary methods' This vector can be obtained from Dr. Solmaz's lab |
| Recombinant DNA reagent | Nup358$^{-min-zip}$ pGEX-6P-1 | GenScript This paper | Modified Nup358$^{-min}$-pGEX-6P-1 vector that includes a leucine zipper that was added at the C-terminus before the snap-tag | For the protein sequence expressed from this vector, see 'Supplementary methods' This vector can be obtained from Dr. Solmaz's lab |
| Recombinant DNA reagent | Human Nup358 (residues 2158–2199) pGEX-6P-1 | GenScript This paper | Sequence encoding residues 2158–2199 of human Nup358 cloned into a pGEX-6P-1 vector via the BamHI and XhoI restriction sites | Protein overexpression plasmid, which can be obtained from Dr. Solmaz's lab |
| Peptide, recombinant protein | PreScission protease | Cytiva | Cat# 27084301 | |
| Peptide, Recombinant protein | Thrombin, human plasma | Fisher Scientific | Cat# 6051951000U | |
| Chemical compound, drug | $^{13}$C D-Glucose (U-13C6) | Cambridge Isotope Laboratories | Cat# CLM-1396-PK | |
| Chemical compound, drug | $^{15}$NH$_4$Cl | Cambridge Isotope Laboratories | Cat# NLM-467-10 | |
| Chemical compound, drug | cOmplete EDTA-free Protease Inhibitor Cocktail tablets | Roche | Cat# 45-5056489001-EA | |
| Software, algorithm | Origin Student 2018 | OriginLab | | CD spectroscopy |
| Software, algorithm | Origin GE Microcal ITC200 | OriginLab | | ITC |
| Software, algorithm | BioXTAS RAW software suite (version 2.0.3). | Reference *Hopkins et al., 2017* | | |
| Software, algorithm | ImageJ 1.52v | Reference *Schneider et al., 2012* | | |
| Software, algorithm | UCSF Chimera (version 1.14) | Resource for Biocomputing, Visualization, and Informatics at the University of California, San Francisco Reference *Pettersen et al., 2004* | | |
| Chemical compound, drug | Coomassie brilliant blue R-250 | VWR | Cat# VWRV0472-25G | |
| Chemical compound, drug | RNase Inhibitor | Promega | N261B | |
| Chemical compound, drug | Q-dot 525 streptavidin conjugate | Invitrogen | Q10141MP | |
| Chemical compound, drug | Q-dot 565 streptavidin conjugate | Invitrogen | Q10131MP | |
| Chemical compound, drug | Q-dot 655 streptavidin conjugate | Invitrogen | Q10121MP | |
| Chemical compound, drug | SNAP-Biotin | New England BioLabs | S9110S | |
| Chemical compound, drug | Tubulin protein (X-rhodamine): bovine brain | Cytoskeleton, Inc | TL620M-A | |
| Chemical compound, drug | Paclitaxel | Cytoskeleton, Inc | TXD01 | |
| Recombinant DNA reagent | Bicaudal D homolog 2 isoform 2 (*Homo sapiens*) | This paper | NCBI:NP_056065.1 | Protein overexpression plasmid, which can be obtained from Dr. Solmaz's lab |
| Biological sample (*Bos taurus*) | Dynein-dynactin | Bovine brain | | |

*Continued on next page*

*Continued*

| Reagent type (species) or resource | Designation | Source or reference | Identifiers | Additional information |
|---|---|---|---|---|
| Biological sample (*B. taurus*) | Tubulin | Bovine brain | | |
| Software, algorithm | Nikon ECLIPSE Ti microscope | Nikon | | |
| Software, algorithm | Nikon NIS Elements | Nikon | | |
| Software, algorithm | Andor EMCCD Camera | Andor Technology USA | | |
| Software, algorithm | Prism | GraphPad | v7; RRID:SCR_002798 | |
| Chemical compound, drug | 2-[Methoxy(polyethyleneoxy)propyl]trimethoxysilane | J&K Scientific | 967192 | |
| Chemical compound, drug | n-Butylamine | Acros Organics | A0344582 | |
| Software, algorithm | ImageJ Fiji | NIH | 1.53c | |
| Software, algorithm | NMRPipe | NIH, reference *Delaglio et al., 1995* | | |
| Software, algorithm | NMRFAM_SPARKY | Reference *Lee et al., 2015* | | |

## Protein expression and purification

Nup358-min and BicD2-CTD were expressed and purified as previously described (*Noell et al., 2019*; *Noell et al., 2018*; *Cui et al., 2020*). For details, see Appendix 3.

## Pull-down assays

GST-pull-down assays of human Nup358-min-GST and human BicD2-CTD were performed as described (*Cui et al., 2020*). For details, see Appendix 3.

## Isothermal titration calorimetry

ITC experiments were performed as previously described (*Noell et al., 2018*). In brief, protein samples were extensively dialyzed against a buffer containing 150 mM NaCl, 30 mM HEPES pH 7.5, 0.5 mM TCEP, and 1 mM $MgCl_2$. For ITC experiments, BicD2-CTD was placed in the cell of a calorimeter (MicroCal Auto-iTC200, GE Healthcare) and titrated with Nup358-min (either without the GST-tag *Figure 1*) or with the GST-tag intact (*Figure 1—figure supplement 1a*) at 25°C. As controls, titrations of Nup358-min (with or without the GST-tag) into buffer were carried out, which resulted in a flat line (*Figure 1—figure supplement 1b and c*). The corrected ITC curves were analyzed using a nonlinear least-squares minimization method in Origin 7.0 and fitted with the one-site model to determine the number of sites N, the equilibrium binding constant K, the change in entropy ΔS, and the change in enthalpy ΔH. The affinity (dissociation constant $K_d$) was calculated as the inverse of K. Data was analyzed with the Origin software (OriginLab). For the titration of Nup358-min + BicD2 CTD, the protein concentrations were 0.27 mM (Nup358-min) and 0.019 mM (BicD2-CTD), respectively. For the titration of Nup358-min-GST + BicD2 CTD, the protein concentrations were 0.12 mM (Nup358-min-GST) and 0.013 mM (BicD2-CTD), respectively. Protein concentrations were determined by UV absorbance at 280 nm.

## Nuclear magnetic resonance

HSQC NMR experiments of $^{15}N$-labeled Nup358-min were recorded on a 0.2 mM, 440 µl sample with 10% $D_2O$ on a Bruker 800 MHz spectrometer equipped with a cryoprobe at 25°C. HSQC was taken on a 1:1 mixture of $^{15}N$-Nup358$^{min}$-BicD2-CTD, where the sample was concentrated to keep the concentration of Nup358-min to 0.2 mM in 20 mM HEPES pH 7.5, 150 mM NaCl, 0.5 mM TCEP. Backbone assignments were accomplished using standard triple resonance experiments and standard NMR processing analysis tools (*Delaglio et al., 1995*; *Ying et al., 2017*; *Lee et al., 2009*; *Lee et al., 2019*). For details, see Appendix 3. $^{15}N$ CEST NMR was initially performed with a 0.6 mM sample and a 10:1 ratio of BicD2-CTD:$^{15}N$-Nup358-min. The temperature was reduced to 20°C, and the pH was reduced to 6.5 to help improve signal/noise by decreasing the rate of solvent exchange of amide protons. $^{15}N$ DCEST NMR (*Yuwen et al., 2018a*; *Yuwen et al., 2018b*) was performed with the same sample conditions. A second sample with a 20:1 ratio of 15N-Nup358-min:BicD2-CTD

was utilized to improve the S/N ratio of the weakest peaks. For both experiments, the 800 MHz spectrometer with cryoprobe was used. For CEST, the saturation pulse was 400 ms at both 10 Hz and 20 Hz. The saturation frequency ranged from 118 ppm (the center of the HSQC spectrum) to 126 ppm in steps of 0.25 ppm in CEST. The second attempt was run from 116 to 124 ppm. The DCEST was performed to cover the whole spectral width for amides, with a 10 Hz saturation and a 700 Hz spectral width, and a 20 Hz saturation pulse with a 600 Hz spectral width. $^{13}$C' HNCO-CEST NMR (*Charlier et al., 2017*) was performed with a 0.6 mM sample and a 1:20 ratio of BicD2-CTD:Nup358-Min at 20°C and pH of 6.5. For this experiment, a 600 MHz spectrometer with cryoprobe was used to minimize the effect of $^{13}$C' transverse relaxation. The saturation pulse was 300 ms at 10 Hz. The saturation frequency ranged from 170 to 180 ppm in steps of 0.33 ppm. The CEST data were fitted with the program RING NMR Dynamics (*Beckwith et al., 2020*) and are presented in *Appendix 1—table 1*.

## CD spectroscopy

CD spectroscopy was performed as previously described (*Cui et al., 2020*). For details, see Appendix 3.

## SAXS experiments

Nup358-min and BicD2-CTD were purified as described above. The monodispersity of the protein was confirmed by SEC-MALS, which is published (*Cui et al., 2019*; *Noell et al., 2018*). Purified Nup358-min and BicD2-CTD were dialyzed against the following buffer: 150 mM NaCl, 20 mM HEPES pH 7.5, 0.5 mM TCEP. The dialysis buffer was used as buffer match for SAXS experiments as well as for dilutions. The following protein concentrations were used: Nup358-min 4 mg/ml, BicD2-CTD 1 mg/ml. To assemble the complex, Nup358-min and BicD2-CTD were mixed in a 1:1 molar ratio and incubated for 30 min on ice, with a final concentration of 1.3 mg BicD2-CTD and 1.3 mg Nup358-min. The affinity of BicD2-CTD towards Nup358-min is 1.7 ± 0.9 μM (*Figure 1*); therefore, this protein concentration is sufficient for complex formation. To assure monodispersity, SAXS data were collected for at least three protein concentrations for each sample, and we also collected data of a Nup358-min/BicD2-CTD complex that was further purified by gel filtration, with comparable results (data not shown). Prior to data collection, samples were thawed, filtered (pore size 0.2 μm), and centrifuged (30 min, 21,700 × *g*, 4°C). SAXS data was collected at the beamline 7A1 at the Cornell High Energy Synchrotron Source, with a dual Pilatus 100k detector system (Dectris, Baden, Switzerland), at a single detector position, on July 3, 2019, as described previously (*Zhao et al., 2021*). Quartz capillary with a path length of 1.6 mm was used as the sample cell (OD = 1.5 mm, wall thickness = 10 μm). For each dataset, 20 frames were collected at 4°C, with 0.1 s exposure times (wavelength = 9.835 keV, beam dimensions = 250 * 250 μm, beam current = 49.9 mA [positrons], beam flux = 2.4 * 10$^{12}$ photons/s). Most samples showed no detectable radiation damage, which was monitored by averaging 20 frames.

SAXS data were processed with the BioXTAS RAW software suite (version 2.0.3) (*Hopkins et al., 2017*). To obtain scattering intensity profiles, 20 data frames were reduced to scattering intensity profiles, placed on an absolute scale, averaged, and the scattering intensity profile of the buffer match was subtracted. The data quality was assessed by Guinier plots, molar mass calculations, and dimensionless Kratky plots in BioXTAS RAW (*Hopkins et al., 2017*; *Hajizadeh et al., 2018*; *Konarev et al., 2003*; *Mylonas and Svergun, 2007*; *Trewhella et al., 2017*). Pair distance distribution p(r) functions were derived from the scattering intensity profiles by the program GNOM (*Svergun, 1992*) of the ATSAS 3.0.0-1 software suite (*Franke et al., 2017*) implemented in RAW (*Hopkins et al., 2017*). Fifteen bead model 3D reconstructions were performed with the Dammiff program (*Franke and Svergun, 2009*) implemented in ATSAS/RAW (*Hopkins et al., 2017*; *Franke et al., 2017*). The resulting models were aligned, grouped into clusters, averaged, and the average model was refined in Dammiff (*Franke and Svergun, 2009*; *Volkov and Svergun, 2003*; *Svergun, 1999*). Figures of the refined molecular envelopes were created in the program UCSF Chimera (version 1.14)(*Pettersen et al., 2004*), developed by the Resource for Biocomputing, Visualization, and Informatics at the University of California, San Francisco, with support from P41-GM103311. P(r) functions were normalized to the highest signal of each curve.

## Protein expression and purification for single-molecule assays

Cytoplasmic dynein and dynactin were purified from 300 g bovine brain as described in *Bingham et al., 1998*, and tubulin was purified from 200 g bovine brain as described in *Castoldi and Popov, 2003*. Purified dynein was stored at −20°C, and dynactin and tubulin were stored at –80°C, in 10 mM imidazole, pH 7.4, 0.2 M NaCl, 1 mM EGTA, 2 mM DTT, 10 µM Mg ATP, 5 µg/ml leupeptin, 50% glycerol. The N-terminal domain of human BicD2 (BicD2$^{CC1}$) with a biotin tag at its N-terminus was expressed in bacteria as described (*Sladewski et al., 2018*). Full-length human wild-type BicD2 was expressed in Sf9 cells as described for *Drosophila* BicD (*Sladewski et al., 2018*). The Bradford reagent (Bio-Rad, USA) was used to measure the protein concentration. To create a fluorescently labeled version of Nup358-min, the expression vector described above was modified to include a SNAP-tag for fluorescent labeling at its CTD, which is referred to as Nup358$^{min}$. Nup358$^{min}$ was expressed and purified with the N-terminal GST-tag intact as described for Nup358-min (*Noell et al., 2018*; *Sckolnick et al., 2013*) using the BL21-DE3-CodonPlus-RIL strain for expression. Nup358$^{min}$ was dimerized using a leucin zipper (hereafter called Nup358$^{min-zip}$); the leucine zipper sequence was added at the C-terminus before the snap-tag. The sequences of these two constructs are shown in Appendix 3. The SNAP tag on Nup358$^{min-zip}$ and Nup358$^{min}$ was biotinylated with SNAP-biotin substrate (New England BioLabs, MA) as described (*Sckolnick et al., 2013*).

## Single-molecule assay

Dynein, dynactin, BicD2, and Nu358$^{min-zip}$ constructs were diluted into high salt buffer (30 mM HEPES pH 7.4, 300 mM potassium acetate, 2 mM magnesium acetate, 1 mM EGTA, 20 mM DTT) and clarified for 20 min at 400,000 × *g* to remove aggregates. To form the dynein-dynactin-BicD2-Nup358$^{min-zip}$ (DDBN$^{min-zip}$) complex, BicD2 and Nup358$^{min-zip}$ were mixed with 525 nm and 655 nm streptavidin Qdots (Invitrogen, CA), respectively, at a 1:1 molar ratio in separate tubes and incubated for 15 min on ice. To block excess binding sites on streptavidin Qdots, 5 µM biotin was added to both tubes. Labeled BicD2 and Nup358$^{min-zip}$ were then mixed with preformed DD complex at a molar ratio of 1:1:2:2 (250 nM dynein, 250 nM dynactin, 500 nM BicD2, and 500 nM Nup358$^{min-zip}$) and incubated on ice for 30 min in motility buffer (30 mM HEPES pH 7.4, 150 mM potassium acetate, 2 mM magnesium acetate, 1 mM EGTA, 20 mM DTT). The dynein-dynactin-Nup358$^{min-zip}$ (DDN$^{min-zip}$) complex contained Nup358$^{min-zip}$ that was labeled with a 655 nm Qdot. In the dynein-dynactin-BicD2$^{CC1}$ (DDB$^{CC1}$) complex, BicD2$^{CC1}$ was labeled with a 525 nm streptavidin Qdot. The DDBN$^{min-zip}$, DDN$^{min-zip}$, and DDB$^{CC1}$ complexes were diluted in motility buffer (30 mM HEPES pH 7.4, 150 mM potassium acetate, 2 mM magnesium acetate, 1 mM EGTA, 2 mM MgATP, 20 mM DTT, 8 mg/ml BSA, 0.5 mg/ml kappa-casein, 0.5% pluronic F68, 10 mM paclitaxel, and an oxygen scavenger system) to a final concentration of 1.25–2.50 nM dynein for observing motion on microtubules. The oxygen-scavenging system consisted of 5.8 mg/ml glucose, 0.045 mg/ml catalase, and 0.067 mg/ml glucose oxidase (Sigma-Aldrich). To analyze the motion of the complexes without any ambiguity, oxygen scavengers were not used in two-color experiments so that the microtubules would photo bleach. Purified tubulin was mixed with rhodamine-labeled tubulin at a molar ratio of 10:1 and polymerized as described (*Sladewski et al., 2018*). PEGylated glass slides were prepared and coated with 0.3 mg/ml rigor kinesin for microtubule attachment as described (*Sladewski et al., 2018*). After rinsing 2–3 times with motility buffer to remove excess rigor kinesin, MTs were added. Excess microtubules were removed by rinsing with motility buffer. Then, dynein protein samples were added to the glass surface. Motion of DDBN$^{min-zip}$, DDN$^{min-zip}$, and DDB$^{CC1}$ were observed using TIRF microscope as described (*Sladewski et al., 2018*).

## Microscopy and data analysis

To detect the motion of Qdot-labeled DDBN$^{min-zip}$ and DDB$^{CC1}$ complexes on microtubules, TIRF microscopy was used. The TIRF microscope system is operated by the Nikon NIS Elements software, and single-molecule images were acquired on a Nikon ECLIPSE Ti microscope equipped with objective-type TIRF. The laser lines 488 and 561 nm were used to illuminate 525 nm and 655 nm Qdots and rhodamine-labeled microtubules. Typically, 300–600 frames were captured at 100 or 200 ms intervals (10 or 5 frames/s) using two Andor EMCCD cameras (Andor Technology USA, South Windsor, CT). Individual Qdots were tracked using the ImageJ MtrackJ plugin for run length and speed measurements of single complexes (*Meijering et al., 2012*). Run length is defined as the total travel distance by individual complexes, and speed was calculated by dividing the run length by the total time. To

determine the characteristic run length, data were plotted as 1-cumulative probability distribution (1-CDF) with GraphPad Prism Software and fit to a one-phase exponential decay equation $p(x) = Ae^{-x/\lambda}$, where $p(x)$ is the relative frequency, $x$ is the travel distance along a microtubule track, and A is the amplitude. Speed was reported as mean ± SD, and run length reported as mean ± standard error (SE). The number of processive events was calculated by counting the number of moving dual-color Qdots per time per µm microtubule. The number of dual-color Qdots (DDBN$^{min-zip}$) was counted on 18–32 microtubules in each case. For measuring the formation of DDBN$^{min-zip}$ and BN$^{min-zip}$ complexes containing WT or mutant Nup358$^{min-zip}$, the number of dual-color and single-color (green or red) Qdots was counted on multiple 512 × 512 pixel fields. The DDBN$^{min-zip}$ and BN$^{min-zip}$ complexes were bound to the glass surfaces nonspecifically. The percentage of colocalization was calculated from the total number of dual-color Qdots divided by all dual- and single-color Qdots. Statistical significance for two sets of run length data was determined by the Kolmogorov–Smirnov test, a nonparametric distribution. For speed data comparison, an unpaired *t*-test was performed. For three or more datasets of run length or speed, statistical significance was calculated using one-way ANOVA followed by Tukey's post-hoc test. Statistical differences for binding frequency of DDBN$^{min-zip}$ containing WT or mutant Nup358$^{min-zip}$ were determined by one-way ANOVA followed by Tukey's post-hoc test.

## Acknowledgements

We thank Fabien Ferrage, Guillaume Bouvignies, and Philippe Pelupessy for help with the CEST experiments and Bruce Johnson for help with fitting of the CEST data. We thank Michael Cosgrove for helpful discussions regarding SAXS. SRS, CW, and MYA were funded by NIH grant R01 GM144578. CW was funded by NIH grants CA206592 and AG069039. SRS was funded by NIH grant R15 GM128119 and additional funds came from the Chemistry Department and the Research Foundation of SUNY. KMT was funded by NIH grant R35 GM136288. MYA was funded by NIH grant R03 NS114115. The CD instrument was supported by NIGMS grants 1R01GM125853-02S1 and 3R35GM130207-01S1. SAXS data was collected at beamline 7A1, Cornell High Energy Synchrotron Source, supported by NSF award DMR-1829070, and by NIH/NIGMS award GM-124166. We thank Qingqiu Huang and Richard Gillilan for user support at the synchrotron source. We also thank Patty Fagnant, Carol Bookwalter, and Elena Krementsova for cloning, protein expression, and protein purification. We thank the High-Throughput and Spectroscopy Resource Center at Rockefeller University for support of the ITC experiments. The authors declare that they have no conflicting interests.

## Additional information

### Funding

| Funder | Grant reference number | Author |
|---|---|---|
| National Institute of General Medical Sciences | R01 GM144578 | M Yusuf Ali<br>Sozanne R Solmaz<br>Chunyu Wang |
| National Cancer Institute | CA206592 | Chunyu Wang |
| National Institute on Aging | AG069039 | Chunyu Wang |
| National Institute of General Medical Sciences | R15 GM128119 | Sozanne R Solmaz |
| Chemistry Department and the Research Foundation of SUNY | | Sozanne R Solmaz |
| National Institute of General Medical Sciences | R35 GM136288 | Kathleen M Trybus |
| National Institute of Neurological Disorders and Stroke | R03 NS114115 | M Yusuf Ali |

| Funder | Grant reference number | Author |
|---|---|---|

The funders had no role in study design, data collection and interpretation, or the decision to submit the work for publication.

## Author contributions

James M Gibson, Data curation, Formal analysis, Investigation, Methodology, Writing - original draft, Writing – review and editing; Heying Cui, Conceptualization, Data curation, Formal analysis, Investigation, Writing - original draft; M Yusuf Ali, Conceptualization, Data curation, Formal analysis, Investigation, Methodology, Writing - original draft, Writing – review and editing; Xiaoxin Zhao, Data curation, Formal analysis, Investigation, Validation; Erik W Debler, Data curation, Formal analysis, Methodology, Writing – review and editing; Jing Zhao, Data curation, Formal analysis, Investigation; Kathleen M Trybus, Conceptualization, Formal analysis, Funding acquisition, Methodology, Resources, Supervision, Validation, Writing - original draft, Writing – review and editing; Sozanne R Solmaz, Conceptualization, Data curation, Formal analysis, Funding acquisition, Investigation, Methodology, Project administration, Resources, Supervision, Validation, Visualization, Writing - original draft, Writing – review and editing; Chunyu Wang, Conceptualization, Formal analysis, Funding acquisition, Investigation, Methodology, Project administration, Resources, Supervision, Writing - original draft, Writing – review and editing

## Author ORCIDs

James M Gibson http://orcid.org/0000-0002-9378-0135
M Yusuf Ali http://orcid.org/0000-0003-2164-3323
Erik W Debler http://orcid.org/0000-0002-2587-2150
Kathleen M Trybus http://orcid.org/0000-0002-5583-8500
Sozanne R Solmaz http://orcid.org/0000-0002-1703-3701
Chunyu Wang http://orcid.org/0000-0001-5165-7959

## Decision letter and Author response

Decision letter https://doi.org/10.7554/eLife.74714.sa1
Author response https://doi.org/10.7554/eLife.74714.sa2

# Additional files

## Supplementary files

• Transparent reporting form

## Data availability

Protein backbone assignments have been deposited in the BMRB under accession code 5182. All other data generated or analyzed during this study are included in the manuscript and supporting files; Source Data files have been provided for Figures 1, 2, 3, 4, 5, 6, 7, and 8.

The following dataset was generated:

| Author(s) | Year | Dataset title | Dataset URL | Database and Identifier |
|---|---|---|---|---|
| Gibson J, Wang C, Zhao J | 2022 | NMR Backbone Assignment of Nup358-Min | https://bmrb.io/data_library/summary/index.php?bmrbId=51282 | Biological Magnetic Resonance Data Bank, 51282 |

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

# Appendix 1

**Appendix 1—table 1.** Summary of the 15N CEST Curve Fits.

**(a) Summary of 15N CEST curve fits at 20:1 molar ratio of BicD2:15N-Nup358**

| Residue | $k_{ex}$ (s–1) | $P_b$ | $\Delta\delta$ (ppm) |
|---|---|---|---|
| A2164 | 704 ± 211 | 0.07 ± 0.02 | –4.3 ± 0.4 |
| K2165 | 139 ± 162 | 0.25 ± 0.08 | –0.8 ± 0.3 |
| L2166 | 381 ± 182 | 0.07 ± 0.03 | –3.8 ± 0.3 |
| I2167 | 975 ± 624 | 0.02 ± 0.04 | –5.7 ± 0.2 |
| K2178 | 31 ± 35 | 0.01 ± 0.01 | –3.6 ± 0.2 |
| L2184 | 160 ± 92 | 0.07 ± 0.03 | –5.5 ± 0.2 |

**(b) Summary of 15N CEST curve fits at 10:1 molar ratio of BicD2:15N-Nup358**

| Residue | $k_{ex}$ (s–1) | $P_b$ | $\Delta\delta$ (ppm) |
|---|---|---|---|
| A2163 | 12 ± 73 | 0.07 ± 0.05 | –3.6 ± 0.2 |
| A2164 | 228 ± 72 | 0.09 ± 0.02 | –3.6 ± 0.2 |
| K2165 | 757 ± 273 | 0.13 ± 0.07 | –3.6 ± 0.2 |
| L2166 | 297 ± 195 | 0.05 ± 0.02 | –3.6 ± 0.2 |

The change in chemical shift in **Table 1** was taken from Table S1b for the alanine residues due to a clearer minor peak. For the others, the change in chemical shift in **Table 1** was taken from Table S1a. The values used in the weighted averages for $k_{ex}$ and $P_b$ were taken from Table S1a, excluding the K2165 and K2178 with abnormal $P_b$ and significant noise in their CEST curves. Including the K2165 and K2178 values, the weighted average $k_{ex}$ value would be 80 ± 30 s$^{-1}$, and the $P_b$ value would be 0.03 ± 0.01.
CEST: chemical exchange saturation transfer.

**Appendix 1—table 2.** Summary of $^{13}$C′ CEST curve fits at 20:1 molar ratio of [BicD2]:[Nup358].

| Residue | $k_{ex}$ (s$^{-1}$) | $P_b$ | $\Delta\delta$ (ppm) |
|---|---|---|---|
| R2162 | 313 ± 328 | 0.05 ± 0.03 | 1.7 ± 0.3 |
| A2164 | 474 ± 304 | 0.07 ± 0.03 | 2.8 ± 0.2 |
| K2165 | 208 ± 179 | 0.05 ± 0.07 | 2.7 ± 0.1 |
| L2166 | 391 ± 170 | 0.04 ± 0.03 | 2.6 ± 0.2 |
| R2169 | 140 ± 125 | 0.06 ± 0.01 | 2.9 ± 0.2 |
| E2171 | 359 ± 175 | 0.07 ± 0.03 | 2.2 ± 0.2 |
| E2172 | 8 ± 98 | 0.22 ± 0.07 | 3.0 ± 0.1 |
| L2177 | 540 ± 241 | 0.03 ± 0.02 | 0.9 ± 0.3 |
| K2181 | 316 ± 155 | 0.06 ± 0.02 | 3.0 ± 0.1 |
| F2183 | 464 ± 98 | 0.06 ± 0.01 | 3.3 ± 0.3 |
| L2184 | 238 ± 165 | 0.08 ± 0.04 | 1.0 ± 0.2 |

## Appendix 2

**Appendix 2—table 1.** Summary of SAXS data.

| Sample | $R_G$ (Å) Guinier plot | $R_G$ (Å) p(r) function | I(0) p(r) function/I(0) Guinier plot | TE / $\chi^2$ p(r) function | $D_{Max}$ p(r) function | MW*[†][‡] (kDa) |
|---|---|---|---|---|---|---|
| Nup358/BicD2 | 48.1 ± 0.6 | 55.5 ± 0.2 | 0.086/0.086 | 0.61/1.04 | 190 | 47.7/46.0/47.6 |
| Nup358-min | 28.4 ± 0.3 | 30.6 ± 0.4 | 0.035/0.036 | 0.69/1.05 | 120 | 12.7/12.3/na |
| BicD2-CTD | 42.6 ± 1.0 | 49.7 ± 0.9 | 0.066/0.070 | 0.67/1.10 | 190 | na/na/38.7 |

Note that not all methods can be applied to all samples (na). The error of molar masses determined by SAXS is 10%. The calculated MWs are 10.9 kDa for BicD2-CTD and 10.6 kDa for Nup358-min.

$R_G$: radius of gyration; I(0): scattering intensity at zero angle; TE: total estimate; $D_{MAX}$: maximum particle diameter; MW: molar mass; SAXS: small-angle X-ray scattering.

*MWs were determined from I(0), using glucose isomerase as a standard (**Konarev et al., 2003**).

[†]MWs were determined from I(0) without a molar mass standard (**Konarev et al., 2003**).

[‡]MWs were determined from the volume of correlation as described by **Rambo and Tainer, 2013**.

**Appendix 2—table 2.** Statistics of SAXS bead model 3D reconstruction.

| Sample | Mean normalized spatial discrepancy score (NSD) | $\chi^2$ (refined model)/ $\chi^2$ (representative model) | Resolution (Å) | Ambimeter score |
|---|---|---|---|---|
| Nup358-min/BicD2-CTD | 0.70 ± 0.05 | 1.08/1.05 | 41 ± 3 | 1.23 |

# Appendix 3

## Supplementary methods
### Sequences of expression constructs
#### Nup358$^{min}$

MW 56.5kDa
MSPILGYWKIKGLVQPTRLLLEYLEEKYEEHLYERDEGDKWRNKKFELGLEFPNLPYYIDGDVKLTQS
MAIIRYIADKHNMLGGCPKERAEISMLEGAVLDIRYGVSRIAYSKDFETLKVDFLSKLPEMLKMFEDR
LCHKTYLNGDHVTHPDFMLYDALDVVLYMDPMCLDAFPKLVCFKKRIEAIPQIDKYLKSSKYIAWPLQ
GWQATFGGgDHPPKSDLEVLFQGPLGSDIPLQTPHKLVDTGRAAKLIQRAEEMKSGLKDFKTFL
TNDQTKVTEEENKGSGTGAAGASDTTIKPNPENTGPTLEWDNYDLREDALDDSVSSMSGDKDCE
MKRTTLDSPLGKLELSGCEQGLHRIIFLGKGTSAADAVEVPAPAAVLGGPEPLMQATAWLNAYF
HQPEAIEEFPVPALHHPVFQQESFTRQVLWKLLKVVKFGEVISYSHLAALAGNPAATAAVKTALSGNP
VPILIPCHRVVQGDLDVGGYEGGLAVKEWLLAHEGHRLGKPGLG

#### Nup358$^{min-zip}$

MW 60.3 kDa
MSPILGYWKIKGLVQPTRLLLEYLEEKYEEHLYERDEGDKWRNKKFELGLEFPNLPYYIDGDVKLTQS
MAIIRYIADKHNMLGGCPKERAEISMLEGAVLDIRYGVSRIAYSKDFETLKVDFLSKLPEMLKMFEDR
LCHKTYLNGDHVTHPDFMLYDALDVVLYMDPMCLDAFPKLVCFKKRIEAIPQIDKYLKSSKYIAWPLQ
GWQATFGGgDHPPKSDLEVLFQGPLGSDIPLQTPHKLVDTGRAAKLIQRAEEMKSGLKDFKTFL
TNDQTKVTEEENKGSGTGAAGASDTTIKPNPENTGPTLEWDNYDLREDALDDSVSSMKQLEDKV
EELLSKNYHLENEVARLKKLVGERMSGDKDCEMKRTTLDSPLGKLELSGCEQGLHRIIFLGKGT
SAADAVEVPAPAAVLGGPEPLMQATAWLNAYFHQPEAIEEFPVPALHHPVFQQESFTRQVLWKL
LKVVKFGEVISYSHLAALAGNPAATAAVKTALSGNPVPILIPCHRVVQGDLDVGGYEGGLAVKE
WLLAHEGHRLGKPGLG

#### Nup358-min

10.5 kDa
GPLGSDIPLQTPHKLVDTGRAAKLIQRAEEMKSGLKDFKTFLTNDQTKVTEEENKGSGTGAAGA
SDTTIKPNPENTGPTLEWDNYDLREDALDDSVSS

#### BicD2-CTD

10.9 kDa
GSHMYENEKAMVTETMMKLRNELKALKEDAATFSSLRAMFATRCDEYITQLDEMQRQLAAAEDE
KKTLNSLLRMAIQQKLALTQRLELLELDHE

## Protein expression and purification

Proteins were expressed and purified as previously described (**Noell et al., 2019**; **Noell et al., 2018**; **Cui et al., 2020**). Codon-optimized expression constructs and point mutants were created by a commercial cloning and gene synthesis service (GenScript). Sequences are listed in Appendix 3. For purification of isotope-labeled Nup358-min, the previously described expression vector for Nup358-min (which encoded the sequence for residues 2148–2240 of human Nup358 cloned into a pGEX6p1 vector) (**Noell et al., 2019**) was expressed in the *Escherichia coli* Rosetta 2(DE3)-pLysS strain as described (**Noell et al., 2019**), with the following modifications: M9 minimal media (3 g $KH_2PO_4$, 6.8 g $Na_2HPO_4$, 1 g NaCl, 4 g D-glucose, 1 g $NH_4Cl$, 2 mM $MgSO_4$, 0.1 mM $CaCl_2$, 100 µg ampicillin, and 35 µg chloramphenicol per liter medium) were used for the expression, which was performed at 37°C. For expression of $^{15}N$-labeled Nup358-min, unlabeled $NH_4Cl$ was replaced in the medium by 1 g $^{15}NH_4Cl$ per liter. For expression of $^{13}C$ and $^{15}N$-labeled Nup358-min, 1 g $^{13}C$ D-glucose (U-13C6) and 1 g $^{15}NH_4Cl$ were used per 1 l of M9 media instead of the unlabeled compounds. Isotopes were obtained from Cambridge Isotope Laboratories.

Nup358-min was purified using the previously described protocol (**Cui et al., 2019**), but the cell pellet from a 6 l bacteria culture was dissolved in 75 ml of the lysis buffer to which 1 ½ tablets of cOmplete EDTA-free Protease Inhibitor Cocktail tablets were added (Roche). In short, Nup358-min was purified by glutathione affinity chromatography and eluted by proteolytic cleavage on the column with PreScission protease. The protein was further purified by size-exclusion chromatography as described (**Cui et al., 2019**), using the following buffer: 20 mM HEPES pH 7.5, 150 mM NaCl, 0.5 mM TCEP.

BicD2-CTD was expressed and purified as previously described (*Noell et al., 2019*; *Cui et al., 2019*; *Noell et al., 2018*). In short, BicD2-CTD was purified by Ni-NTA affinity chromatography, followed by proteolytic cleavage of the His$_6$-tag by thrombin. BicD2-CTD was further purified by a second round of Ni-NTA affinity chromatography, followed by size-exclusion chromatography in the same buffer as described above. The Nup358 (residues 2158–2199)/BicD2-CTD complex was purified as described for the Nup358-min/BicD2-CTD complex (*Noell et al., 2019*).

## Pull-down assays

GST-pull-down assays of human Nup358-min-GST and human BicD2-CTD were performed as described (*Cui et al., 2020*). For the assays, Nup358-min-GST was immobilized on glutathione sepharose beads and incubated with purified BicD2-CTD prior to elution with glutathione. The elution fractions were analyzed by SDS-PAGE (16% acrylamide gels) and stained with Coomassie blue. ImageJ was used for quantification of gel bands (*Schneider et al., 2012*).

## Nuclear magnetic resonance

For backbone assignment, triple resonance experiments (HNCO, HNCA, HNCACO, HNCOCA, HNCACB, CBCCACONH, and HNN; *Panchal et al., 2001*) were performed on double-labeled $^{15}$N/$^{13}$C Nup358-min at 0.4 mM on a Bruker 800 MHz spectrometer equipped with a cryoprobe. They were performed with nonuniform sampling (NUS). Processing of the data was performed with NMRPipe (*Delaglio et al., 1995*) and SMILE (*Ying et al., 2017*). Further analysis of the data was performed with NMRFAM_SPARKY (*Lee et al., 2015*), including iPine (*Lee et al., 2009*; *Lee et al., 2019*).

## CD spectroscopy

Purified proteins at a concentration of 0.3 mg/ml were dialyzed in the following buffer: 150 mM NaCl, 10 mM Tris pH 8, and 0.2 mM TCEP. Data were recorded with a Jasco J-1100 CD Spectrometer, equipped with a thermoelectric control device. A quartz cuvette with a path length of 0.1 cm was used. After the buffer baseline subtraction, CD signals were normalized to the protein concentration and converted to mean residue molar ellipticity $[\Theta]$. Thus, the ellipticity $\Theta$ (mdeg) was multiplied with the conversion factor 391.51 cm$^2$/dmol for all spectra with the exception of the Nup358-min spectrum, for which the conversion factor was 362.00 cm$^2$/dmol. For the Nup358 + BicD2 spectrum, the spectra of Nup358-min and BicD2-CTD were added together prior to conversion to $[\Theta]$.

