## [Editor Report]

The paper provides evidence that a peptide from the nuclear pore protein Nup358 (RanBP2) forms a helix upon binding the dynein/dynactin adaptor BICD2 and is able to activate BICD2 in the same way as the small G-protein Rab6. Helix formation to activate BicD2 is a novel concept and will open up the search space for other potential activators that may be unstructured in their apo states.

---

## [Decision Letter]

**Decision letter after peer review:**

[Editors’ note: the authors submitted for reconsideration following the decision after peer review. What follows is the decision letter after the first round of review.]

Thank you for submitting the paper "Coil-to-Helix Transition at the Nup358-BicD2 Interface Activates BicD2 for Dynein Recruitment" for consideration by *eLife*. Your article has been reviewed by 4 peer reviewers, one of whom is a member of our Board of Reviewing Editors, and the evaluation has been overseen by a Senior Editor. The following individual involved in review of your submission has agreed to reveal their identity: Elisar J Barbar (Reviewer #3).

We are sorry to say that, after consultation with the reviewers, we have cannot invite revisions of the manuscript at this stage. The reviewers agreed that the main finding of the paper is novel and potentially of interest to *eLife*'s readership, i.e. that a peptide from Nup358 can form a helix that activates BicD2 in the same way as the small G-protein Rab6. However, at this stage they had a number of experimental concerns including: the need for control TIRF experiments to show a direct comparison of the effect on dynein motility of BICD2 alone and BICD2+Nup358, negative controls for ITC experiments and some extra NMR experiments. Addressing these issues would be required if you want to submit the paper again to *eLife*. In addition, the reviewers all felt that the structural aspects of the paper would be strengthened by data mapping where on BICD2 the Nup358 binds.

*Reviewer #1:*

In this manuscript, Gibson et al. explore the mechanisms underpinning Nup358 recognition by the dynein adaptor BicD2 using a combination of structural and biochemical techniques. Nup358 is a nuclear pore protein and has previously been shown to recruit the dynein machinery via BicD2 for nuclear positioning during G2 phase. The authors perform elegant in vitro motility assays to demonstrate that binding to Nup358 activates BicD2 for dynein recruitment. They also show that the interaction between Nup358 and BicD2 is enhanced in the presence of dynein and dynactin, which is consistent with published models of BicD2 activation. They then move on to investigate the structural basis for the Nup358-BicD2 interaction using multiple structural and biophysical methods, including NMR spectroscopy. They find that Nup358 contains an intrinsically disordered region (IDR) which undergoes a coil-to-helix transition upon binding to BicD2. This observation is supported by mutagenesis experiments which highlight the importance of helix interface residue in BicD2 binding and dynein activation. Interestingly, the authors show that Nup358 harbours non-overlapping binding sites for BicD2 and kinesin-1, hinting that opposite polarity motors may be simultaneously recruited to the nuclear pore for nuclear positioning. These data provide important mechanistic insight on how BicD2 recognizes its cargo for dynein-based transport and represents a significant contribution to the field. While the experiments described in this manuscript are overall convincing, there are a few concerns regarding the in vitro motility studies.

1) It is not clear how the authors are quantifying the percentage of DDBN complexes in Figures 2b-d. Is this solely based on the colocalization of Nup358 and BicD2 or are other factors taken to account (e.g. binding to microtubules, motility)? If the former is true, this is not the most elegant way to prove that all the components are coming together, since the authors have shown that Nup358 and BicD2 can form a complex independently of dynein and dynactin (although with lower affinity). A better way of quantifying DDBN complexes is showing the percentage of complexes that contain both of Nup358 and BicD2 and are motile rather than static.

2) Figure 2 shows in vitro motility data for DDBCC1, DDNmin-zip and DDBNmin-zip complexes, but not for dynein/dynactin/full-length BicD2 (DDB) complexes in the absence of Nup358min-zip. The latter control is critical to be able to accurately quantify the activating effect of Nup358min-zip on BicD2 and needs to be included in the final version of the manuscript. We refer the authors to previous work by Huynh and Vale (2017) which suggests that DDB complexes are not fully static but exhibit some background motility. A DDB control is also important to interpret the differences in speed and run length shown in Figures 2h-2i.

3) In the introduction, the expression dynein adaptor is used to describe proteins that activate dynein motility, while cargo adaptor refers to proteins that bind dynein adaptors to connect them to cargoes. However, in the literature the terms dynein adaptor and cargo adaptor are often used interchangeably to describe activating adaptors such as Hook3 and Bicd2. Therefore, the distinction made by the authors could be confusing for readers in the field. We suggest that the authors use the term adaptor-binding protein to refer to Nup358, as well as Rab6 and nesprin 2-G.

4) In the introduction, the discussion on the LIC1 and BICD2 interaction could be shortened, as it is not directly relevant to the scope of the paper.

5) It would be helpful to show the data in support of the author's statement: "unlike the DDNmin-zip complex, Nup358min-zip alone does not bind to microtubules"

7) The findings by Huynh and Vale (2017), which show BICD2 activation by Rab6 in TIRF motility assays, should be cited and discussed by the authors.

8) In Figure 8, the authors should show the percentage of motile complexes for DDBN with wild-type and mutant Nup358min.

9) In the discussion, the authors should briefly discuss the evidence in support of their statement "It is conceivable that the cargo adaptors Rab6GTP and nesprin-2G are also recognized by similar short α-helices, although this needs to be confirmed by future experiments". For example, are the BICD2-binding domains of Rab6 and/or nesprin 2-G known to contain α-helices or IDRs that could transition into α-helices upon interaction with BICD2?

10) The authors should clarify and more critically discuss their finding that "Nup358min-zip binds BicD2 alone poorly". The readout for the mutagenesis experiment shown in Figure 7 is based on the interaction between Nup358min-zip and BICD2, which seems convincingly solid. Furthermore, the dissociation constant shown in Figure 1d suggests a moderate affinity.

11) The findings from Liu et al. (2013), which shed light on the mechanisms of BicD cargo recognition and autoinhibition, are relevant to this study and should be cited and discussed in the introduction and/or discussion.

*Reviewer #2:*

The authors investigate the interaction of the nuclear pore protein Nup358 with the dynein-dynactin-BicD2 transport complex and localize the interaction site on Nup358 to a short region that transitions between a random coil conformation in the unbound state to an α-helix in the bound state. The residues on one side of the helix that interact with BicD2 are mapped both through structural and functional assays. This work expands our knowledge of how a cargo protein, Nup358 on the nucleus, activates a dynein activating protein, BicD2, which is normally in an inhibited state when not bound to cargo.

The strength of this work is that a number of biochemical, structural, and functional assays are used to pick apart the details of an important interaction between a cargo adapter protein, Nup358 and a dynein activating protein, BicD2. It is known that activator proteins like BicD2, hook, spindly, and others are necessary to fully activate the dynein-dynactin complex. BicD2 is itself regulated, however, because it normally exists in a folded and auto-inhibited form. Thus, understanding cargo transport requires understanding both what are the cargo adapters, as well as how they disinhibit the dynein activators to turn on motility. The interesting part of the current study is that the domain on Nup358 transitions from a random coil conformation in the apo state to an α helix when bound to BicD2. This transition and the subsequent tight binding is what transitions BicD2 out of its autoinhibited conformation, and allows it to activate dynein. It is nice to fully document this interaction, as the details of dynein activation and cargo interaction are still being discovered, and it is possible that this disordered to ordered transition is a common feature. The data are laid out well for the most part and detailed extensively. The text is somewhat verbose, and the authors emphasize the techniques quite a lot at the expense of clearly laying out the questions and hypotheses. Also, the cell biological impact of what we learn about bidirectional cargo transport by dynein and kinesin is limited, as this is more of a technical paper detailing a structural interaction.

The authors bring together a nice suite of tools to study a very focused question regarding the interaction of the cargo binding protein Nup358 with the dynein activating protein BicD2. That these proteins interact, and that this interaction underlies nuclear transport by dynein has been established, and this current work focuses on the details of this interaction. The focus and detail of the paper were strengths, as was the combination of structure, biochemistry, and functional assays. The work doesn't challenge any paradigms and in the end is much more of a biochemical investigation than one that impact cell biology. That would be just fine for JBC, but limits the general audience, which is generally what is expected of an *eLife* paper. I found only minimal technical problems, which are documented below. I think the paper could be improved by more focused writing, as well as by a a great emphasis of the specific questions being asked and the broader implications of the work.

Specific Comments:

I appreciated the thorough background that the introduction provided. However, the authors should do a better job of defining the specific questions being addressed here, and the specific things not known that this study is addressing. Also, I thought the last paragraph was too much advertising (e.g. interdisciplinary) and emphasized the "how" of the study rather than the "what did we learn".

Since one key point is to show that Nup358 min and min-zip activate DDB, then a negative control showing that complex is not active in absence of Nup358 is important to show. This would complement the positive control of BicD2-cc.

For the Figure 2 data, please define "% complex formation" more clearly; also, there is no need to give numbers for % complex in legend (it's in the text). For 2i, I suggest a 1-cdf plot and fit the exponential to that, which eliminates problems with binning.

The first paragraph of discussion is much too excessive summary of the findings – it should be distilled to the results, not a recap of the methods and technical details.

I know that acronyms streamline things, but they put off readers who often have to go back and check what they mean. It seems like there must be a better way than a sentence like on page 19: "In the BN complex, the NTD of BicD2 competes with Nup358 for binding to the CTD, likely resulting in a lower apparent affinity than in the DDBN complex, where the NTD interacts with dynein and dynactin and prevents reassociation of the NTD with the CTD."

*Reviewer #3:*

A summary of what the authors were trying to achieve.

– Identify the molecular details of the Nup358/BicD2 interaction by determining the minimum Nup358 domain needed for binding

– Determine a structural mechanism of BicD2-mediated cargo recognition

– Provide mechanistic insight on dynein activation

An account of the major strengths and weaknesses of the methods and results.

Strengths:

– The combination of techniques and organization of results is done well and is easy to follow

– Emphasizing the larger picture of a dynein/dynactin/BicD2/Nup358 complex is impactful and helpful to readers to keep in mind the bigger picture implications of the work.

– Techniques are well explained as far as data interpretation

Weaknesses:

– Some of the data (ITC & CEST) are fit to a simple model without much discussion about that the binding involves a coiled coil dimer.

– The random coil to helix transition would be better supported by CA & CB chemical shifts. With assignments in hand the chemical shifts for apo and bound could be graphed to highlight secondary structure propensity.

– Would be helpful to show the secondary structure prediction mentioned on page 14, what is the structure of the free protein, is there helical prediction or propensity in the region of the coil that transitions to helix?

– A discussion of what did we learn from the kinetics will be helpful. The kinetics of coil to helix are well reported in the literature. How are these similar?

An appraisal of whether the authors achieved their aims, and whether the results support their conclusions.

– Authors do appear to achieve their aims in determining the binding site on Nup358 for BicD2 while also proposing a mechanism for BicD2 cargo recognition

– Suggested/implied conclusions leave the authors plenty of future work especially in regards to BicD2 regulation and possible bidirectional Nup complexes

– The random coil to helix transition of Nup would be better supported by additional NMR and CEST data (mentioned above), however the coupling of CD data and secondary structure prediction (although not shown) do strengthen the conclusion

– In the mutagenesis studies, particularly in Fig7a, the differences between some residues (2169, 2172, 2181) seem too small for the conclusions made.

A discussion of the likely impact of the work on the field, and the utility of the methods and data to the community.

– Impact of the work is well summarized in the discussion/conclusion as these findings may be universal to many more dynein/cargo adaptor pairs

– Methodology seems standard, although the future directions in regards to BicD2 regulation and possible bidirectional Nup complexes are interesting.

– Any additional context you think would help readers interpret or understand the significance of the work.

– Authors could discuss how/why this work applies to complexes that are not specific to dynein (other instances of autoinhibition regulation)

– The repeated claim of a 20:1 (BicD2:Nup358) molar ratio for the CEST experiments is confusing, perhaps the authors mean 1:20? This should be corrected in multiple places.

– The NMR titration is not a true titration as only a 1:1 molar ratio is shown. It is also unclear if peak intensities are taken from peak height or volume and a discussion of error bar calculation is needed to convince of the significance of the measurements.

– To say that "Nup358 relieves the auto-inhibition of BicD2 so that dynein-dynactin can bind and move processivity on MTs" would perhaps be better stated as "functionally relieves auto-inhibition" since results are clearly not the same as when BicD2 is truncated and auto-inhibition is truly absent. (page 7)

*Reviewer #4:*

In this manuscript by Gibson and colleagues provides new insights into the structural and molecular basis for Nup358 binding to the dynein activating adaptor BicD2 and its implications in dynein's motility. In line with their previous publication (Noell et al., 2018), the authors show that the small fragment in Nup358 drives binding to BicD2. The authors quantitatively characterize the interaction using isothermal titration calorimetry (ITC) and confirm a binding stoichiometry of a Nup358 dimer to a BicD2 dimer. By using a combination of NMR spectroscopy, CD spectroscopy, pull-downs and SAXS experiments, they demonstrate that BicD2 recognizes Nup358 through a short "cargo recognition α-helix", which transitioned from a random coil to an α-helix upon BicD2 binding. Finally, the authors present evidence that the interaction between BicD2 and Nup358 is important for the formation of dynein/dynactin/BicD2/Nup358 motor complex, as well as the processive motility of the motor complex in vitro. Overall, the conclusions are well supported by the data and the findings clarify previously reported results. The solution NMR and in vitro motility data add significantly to the understanding of how cargo binding to activating adaptors may promote the processive motility of the motor complex.

Pending points below, I think this manuscript merits publication in *eLife*.

1. ITC experiments.

a) In ITC experiments, the titration of protein (placed in the syringe) into the buffer (placed in the calorimetric cell) is lacking as a negative control. The control experiment is particularly important as the protein placed in the syringe is expected to be dimeric in solution (I assume Nup358 is placed in the syringe). The stepwise addition of the solution in the syringe into the cell might trigger the dissociation reaction of the Nup358 homodimer, leading to the detection of the heat.

b) Please provide the experimental details on the ITC experiments.

c) Please provide the concentrations of the proteins used in ITC experiments in the cell and the syringe.

2. NMR experiments.

a) HSQC titration experiments (Figure 3 and Figure S2) are not very convincing as there are several mis-labeled (T2159 in Figure 3, A2440, T2143, G2146 in Figure S2 and so on) or unlabeled peaks. Also, the overall figure quality should be improved (figure resolution, tick marks, residue labels etc.).

b) I highly suggest the authors perform at least one more titration experiment with a different molar ratio (e.g. [Nup358-min]:[BicD2-CTD]=1:0.5), to see whether the decrease of the peak intensities are truly dependent on BicD2 binding.

c) It would be nice if the authors analyze the secondary structure elements using NMR chemical shifts (using TALOS+ for example) and compare the results with the helical propensity detected by CD experiments.

d) The assignment of apo Nup358-min should be deposited in the BMRB database.

3. Single molecule binding and processivity assays.

a) It is surprising that DDBNmin-zip complex formed with the Nup358min-zip-L2177A behaves similarly to WT-Nup358min-zip. Mutagenesis of L2177A impairs the interaction between Nup358 and BicD2 (Figure S12a), and L2177 residue lies at the center of the predicted binding interface (Figure 7c). Could the authors comment on this?

[Editors’ note: further revisions were suggested prior to acceptance, as described below.]

Thank you for submitting your article "Coil-to-Helix Transition at the Nup358-BicD2 Interface Activates BicD2 for Dynein Recruitment" for consideration by *eLife*. Your article has been reviewed by 3 peer reviewers, one of whom is a member of our Board of Reviewing Editors, and the evaluation has been overseen by Anna Akhmanova as the Senior Editor. The reviewers have opted to remain anonymous.

Essential revisions:

1) Perform replicates of the single molecule experiments showing activation of BICD2 by Nup358 and make the raw data available (please see reviewer 1)

2) Tone down some of the conclusions from the NMR work (reviewer 2)

*Reviewer #1:*

1) Regarding Figure 2f, did the authors perform 3 replicates for each condition? This is not specified in the text and is not deducible from the source data provided. To make the claim that Nup358 activates DDB, it is necessary to show that DDBN consistently displays higher motility than DDB in 3 replicates (for each condition). Also, the source data provided by the authors for this experiment is somewhat unhelpful to the reviewers. For the sake of transparency, the authors should provide access to the source movies for this experiment.

2) The second sentence in the abstract "Nup358 binds and activates the auto-inhibited dynein adaptor Bicaudal D2 (BicD2), which in turn recruits and activates the dynein machinery to position the nucleus" implies that activation of BICD2 by Nup358 had been shown previously. To my knowledge, direct activation of BICD2 by Nup358 had not been formally shown before (although we knew that Nup358 recruits dynein via BICD2). The ability of Nup358 to relieve BICD2 from its autoinhibited state in in vitro assays is one of the key discoveries of this manuscript and should be emphasized in the abstract/introduction.

3) The introduction is poorly written and could be improved.

– The first paragraph of the introduction could be rearranged to better emphasize what is known (i.e. that cargo binding to the CTD frees up the NTD for interaction with dynein/dynactin) and what is unknown (i.e. what are the structural basis of cargo recognition and how does this lead to structural changes in BICD2 conformation). For example, the authors first state that "Cargoes are required to activate BicD2 for dynein binding, which is a key regulatory step for dynein-dependent transport but the underlying mechanism is unknown" but later provide a detailed explanation of how BICD2 is autoinhibited "in the absence of cargoes, BicD2 assumes a looped, auto-inhibited conformation, in which its N- terminal dynein/dynactin binding site binds to the CTD and remains inaccessible. The CTD is required for auto-inhibition, as a truncated BicD2 without the CTD activates dynein/dynactin for processive motility. Binding of dynein adaptors/cargo to the CTD releases auto-inhibition, likely resulting in an extended conformation", which is somewhat confusing.

– The sentence "Finally, Rab6GTP recruits BicD2/dynein for the transport of Golgi-derived secretory vesicles" does not fit well with the rest of the paragraph in its current form. Something along the lines "Apart from its roles in nuclear positioning, BicD2 is also involved in the transport of Golgi-derived secretory vesicles. In this process, BicD2 and dynein are recruited by the vesicle-associated small GTPase Rab6" would be more helpful to the reader.

– The paragraph on the importance of IDRs is not very informative, since the idea of an IDR that transitions into a cargo-recognition helix has not been introduced at this stage. In my opinion, this paragraph could be removed and replaced by a short sentence in the discussion which highlights the importance of IDRs in dynein biology. If the authors wish to keep it, they should provide more context on why IDRs are relevant to the scope of this research.

*Reviewer #2:*

The discussion of the ITC fits was improved and is clear now.

The explanation for the absence of CA and CB chemical shifts is not, however. A concentration of 0.6 mM should be sufficient for assignments of the free protein. It is not clear to me what experimental chemical shift the author is referring to that do not show stable helical conformation. Were these published before, and if so they will be better included here in the figure instead of prediction. A nascent helix will be sufficient to show.

Also, it is concerning to me that the timescale measured by CEST is so different from coil to helix transition, but still the interpretation of coil to helix transition as the driver has not changed. If CEST is measuring a different process than coil to helix, and there is no strong structural evidence that there is a helix being formed or has tendency to form, then the interpretation of coil to helix transition is not supported.

*Reviewer #3:*

This manuscript by Gibson and colleagues is a resubmission version of a previously rejected paper (26-07-2021-RA-*eLife*-72507). The authors have satisfactorily addressed all of my previous concerns in the resubmitted manuscript. The inclusion of the appropriate control experiments as well as NMR titration data strengthens the conclusions.

---

## [Author Response]

[Editors’ note: the authors resubmitted a revised version of the paper for consideration. What follows is the authors’ response to the first round of review.]

Reviewer #1:In this manuscript, Gibson et al. explore the mechanisms underpinning Nup358 recognition by the dynein adaptor BicD2 using a combination of structural and biochemical techniques. Nup358 is a nuclear pore protein and has previously been shown to recruit the dynein machinery via BicD2 for nuclear positioning during G2 phase. The authors perform elegant in vitro motility assays to demonstrate that binding to Nup358 activates BicD2 for dynein recruitment. They also show that the interaction between Nup358 and BicD2 is enhanced in the presence of dynein and dynactin, which is consistent with published models of BicD2 activation. They then move on to investigate the structural basis for the Nup358-BicD2 interaction using multiple structural and biophysical methods, including NMR spectroscopy. They find that Nup358 contains an intrinsically disordered region (IDR) which undergoes a coil-to-helix transition upon binding to BicD2. This observation is supported by mutagenesis experiments which highlight the importance of helix interface residue in BicD2 binding and dynein activation. Interestingly, the authors show that Nup358 harbours non-overlapping binding sites for BicD2 and kinesin-1, hinting that opposite polarity motors may be simultaneously recruited to the nuclear pore for nuclear positioning. These data provide important mechanistic insight on how BicD2 recognizes its cargo for dynein-based transport and represents a significant contribution to the field. While the experiments described in this manuscript are overall convincing, there are a few concerns regarding the in vitro motility studies.1) It is not clear how the authors are quantifying the percentage of DDBN complexes in Figures 2b-d. Is this solely based on the colocalization of Nup358 and BicD2 or are other factors taken to account (e.g. binding to microtubules, motility)? If the former is true, this is not the most elegant way to prove that all the components are coming together, since the authors have shown that Nup358 and BicD2 can form a complex independently of dynein and dynactin (although with lower affinity). A better way of quantifying DDBN complexes is showing the percentage of complexes that contain both of Nup358 and BicD2 and are motile rather than static.

As the reviewer suggested, we have now quantified the number of moving complexes (processive events per min per micrometer MT) of DDBN, and as controls DDB^CC1^, DDB and DDN in Figure 2f.

2) Figure 2 shows in vitro motility data for DDBCC1, DDNmin-zip and DDBNmin-zip complexes, but not for dynein/dynactin/full-length BicD2 (DDB) complexes in the absence of Nup358min-zip. The latter control is critical to be able to accurately quantify the activating effect of Nup358min-zip on BicD2 and needs to be included in the final version of the manuscript. We refer the authors to previous work by Huynh and Vale (2017) which suggests that DDB complexes are not fully static but exhibit some background motility. A DDB control is also important to interpret the differences in speed and run length shown in Figures 2h-2i.

We have quantified the number of processive events of DDB which is ~10 times lower than that of DDBN. Because the number of processive events is so small (which is the key observation), we did not display the speed and run length of these infrequent events in Figure 2g or 2h. We have only 7 data points for DDB. The mean speed and run length of these data were 0.16 ± 0.08 µm/s and 1.5±0.3 µm.

3) In the introduction, the expression dynein adaptor is used to describe proteins that activate dynein motility, while cargo adaptor refers to proteins that bind dynein adaptors to connect them to cargoes. However, in the literature the terms dynein adaptor and cargo adaptor are often used interchangeably to describe activating adaptors such as Hook3 and Bicd2. Therefore, the distinction made by the authors could be confusing for readers in the field. We suggest that the authors use the term adaptor-binding protein to refer to Nup358, as well as Rab6 and nesprin 2-G.

We replaced the term cargo adapter with the terms adapter-binding protein, cargo or interacting partner (depending on the context) throughout the manuscript.

4) In the introduction, the discussion on the LIC1 and BICD2 interaction could be shortened, as it is not directly relevant to the scope of the paper.

We addressed that point by shortening the description in the introduction.

5) It would be helpful to show the data in support of the author's statement: "unlike the DDNmin-zip complex, Nup358min-zip alone does not bind to microtubules"

We show in Figure S2(d) that the Nup358^min-zip^ alone does not bind to microtubules.

7) The findings by Huynh and Vale (2017), which show BICD2 activation by Rab6 in TIRF motility assays, should be cited and discussed by the authors.

The findings of Huynh and Vale (2017) is now included in the Discussion (first paragraph):

“Single-molecule processivity assays revealed that a dimerized Nup358-min domain forms a complex with dynein/dynactin/BicD2 (DDBN) that shows robust processive motility on microtubules. Similarly, the small GTPase Rab6a^GTP^ that links BicD2 to membranous cargo promotes processive motility (Huynh and Vale JCB 2017), suggesting that both adaptor-binding proteins act to release the auto-inhibition of BicD2.”

8) In Figure 8, the authors should show the percentage of motile complexes for DDBN with wild-type and mutant Nup358min.

We have quantified moving processive events for DDBN and mutants in Figure 8a.

9) In the discussion, the authors should briefly discuss the evidence in support of their statement "It is conceivable that the cargo adaptors Rab6GTP and nesprin-2G are also recognized by similar short α-helices, although this needs to be confirmed by future experiments". For example, are the BICD2-binding domains of Rab6 and/or nesprin 2-G known to contain α-helices or IDRs that could transition into α-helices upon interaction with BICD2?

We added this point to the discussion.

“Rab6GTP binds to BicD2 with 10-fold higher affinity compared to Rab6GDP, and associated structural changes are found in the Switch I and Switch 2 regions of Rab6, which include short α-helices and intrinsically disordered domains and are possibly candidates for the BicD2-binding site. The mapped nesprin-2G binding domain for BicD2 is also predicted to be to a large degree intrinsically disordered. ”

10) The authors should clarify and more critically discuss their finding that "Nup358min-zip binds BicD2 alone poorly". The readout for the mutagenesis experiment shown in Figure 7 is based on the interaction between Nup358min-zip and BICD2, which seems convincingly solid. Furthermore, the dissociation constant shown in Figure 1d suggests a moderate affinity.

We agree that the term “poorly” is misleading, and we edited this sentence accordingly:

“Nup358min-zip binds to BicD2 alone with lower affinity compared to the complex that also contained dynein and dynactin, and the percentage of BN was reduced for four Nup358min-zip mutants (Figure 8b).”

11) The findings from Liu et al. (2013), which shed light on the mechanisms of BicD cargo recognition and autoinhibition, are relevant to this study and should be cited and discussed in the introduction and/or discussion.

We agree. This paper is cited in the introduction three times for the BicD cargo recognition and autoinhibition mechanism and four times in the discussion.

Reviewer #2:[…]Specific Comments:I appreciated the thorough background that the introduction provided. However, the authors should do a better job of defining the specific questions being addressed here, and the specific things not known that this study is addressing. Also, I thought the last paragraph was too much advertising (e.g. interdisciplinary) and emphasized the "how" of the study rather than the "what did we learn".

The introduction was edited to address this comment. Specific questions addressed by our study were carved out throughout the introduction and the last paragraph of introduction was edited to emphasize “what we learned” from the study:

“Here we have determined the structural properties of the interface of a minimal Nup358/BicD2 complex by a combination of NMR spectroscopy, mutagenesis, circular dichroism spectroscopy and small-angle X-ray scattering. These results establish a structural basis for cargo recognition by BicD2 and suggest that Nup358 interacts with BicD2 through a “cargo recognition” α-helix. This work is further enhanced by singlemolecule binding and processivity assays, which show that the minimal dimerized Nup358 construct is sufficient to activate full-length dynein/dynactin/BicD2 complexes for processive motility. Mutations of the cargo-recognition α-helix of Nup358 decreased the Nup358/BicD2 interaction and resulted in decreased dynein recruitment and impaired motility, shedding light on the important role of the cargo-recognition α-helix for the activation of BicD2 and dynein motility. Intriguingly, our results also show that the binding site of BicD2 in Nup358 is spatially close to but does not overlap with the LEWD motif that acts as a kinesin-1 binding site, suggesting that the kinesin and dynein machineries may interact simultaneously via Nup358. Our results thus provide mechanistic insights into the regulation of bi-directional transport by adapter binding proteins.”

Since one key point is to show that Nup358 min and min-zip activate DDB, then a negative control showing that complex is not active in absence of Nup358 is important to show. This would complement the positive control of BicD2-cc.

DDB control is shown in Figure 2f, demonstrating much reduced motility in the absence of Nup358.

For the Figure 2 data, please define "% complex formation" more clearly; also, there is no need to give numbers for % complex in legend (it's in the text). For 2i, I suggest a 1-cdf plot and fit the exponential to that, which eliminates problems with binning.

We have defined complex formation in the method section:

“For measuring the formation of DDBN^min-zip^ and BN^min-zip^ complexes containing WT or mutant Nup358^min-zip^, the number of dual color and single color (green or red) Qdots were counted on multiple 512 x 512 pixel fields. The DDBN^min-zip^ and BN^min-zip^ complexes were bound to the glass surfaces non-specifically. The percentage of colocalization was calculated from the total number of dual color Qdots divided by all dual and single color Qdots. “

The first paragraph of discussion is much too excessive summary of the findings – it should be distilled to the results, not a recap of the methods and technical details.

The first paragraph of the discussion was shortened accordingly.

I know that acronyms streamline things, but they put off readers who often have to go back and check what they mean. It seems like there must be a better way than a sentence like on page 19: "In the BN complex, the NTD of BicD2 competes with Nup358 for binding to the CTD, likely resulting in a lower apparent affinity than in the DDBN complex, where the NTD interacts with dynein and dynactin and prevents reassociation of the NTD with the CTD."

This sentence was edited and we reviewed the discussion to reduce acronyms. The edited sentence is below:

“In the full-length dynein/dynactin/BicD2/Nup358 (DDBN) complex, once Nup358 binding abolishes the auto-inhibited loop in BicD2, dynein and dynactin bind to BicD2 and prevent the re-establishment of the auto-inhibition. This is not true in the full-length BicD2/Nup358 complex, resulting in a lower apparent affinity than in the full complex.”

Reviewer #3:A summary of what the authors were trying to achieve.– Identify the molecular details of the Nup358/BicD2 interaction by determining the minimum Nup358 domain needed for binding– Determine a structural mechanism of BicD2-mediated cargo recognition– Provide mechanistic insight on dynein activationAn account of the major strengths and weaknesses of the methods and results.Strengths:– The combination of techniques and organization of results is done well and is easy to follow.– Emphasizing the larger picture of a dynein/dynactin/BicD2/Nup358 complex is impactful and helpful to readers to keep in mind the bigger picture implications of the work.– Techniques are well explained as far as data interpretationWeaknesses:– Some of the data (ITC & CEST) are fit to a simple model without much discussion about that the binding involves a coiled coil dimer.

We did fit more complex models to the ITC thermogram, however, these more complex models are not supported by statistical analysis. Thus, to address this point, we improved the description of the ITC data analysis. Based on the ITC analysis, we did not feel it was necessary to fit the ITC or CEST to a more complex model.

Revised Results section:

The ITC thermogram fits well to a one-site binding model (with a single equilibrium dissociation constant K_D_). The number of sites was determined to be N = 1.0, which is consistent with a molar ratio of [Nup358]/[BicD2] of 1. This molar ratio is in agreement with our previously published molar masses obtained from size exclusion chromatography coupled to multi-angle light scattering (SEC-MALS), which showed that Nup358 and BicD2 form a 2:2 complex^50^, while apo-BicD2 forms a dimer^50^ and apoNup358-min forms a monomer. The ITC thermogram, the one-site binding model and the 2:2 stoichiometry are in line with Nup358-min forming a dimer as an intermediate that then binds to a single binding site on a BicD2 coiled-coil dimer, although we cannot exclude the possibility that two Nup358 monomers bind to two binding sites on BicD2, where both sites have the same dissociation constant K_D_. We also attempted to fit models assuming multiple binding sites and K_Ds_ but those are not supported by the ITC thermogram.

Revised Discussion section:

The affinity of the monomeric cargo adaptor Nup358-min for BicD2 is enhanced by dimerization by a leucine zipper, similar to what was previously observed for binding of the cargo adaptor Egalitarian for *Drosophila* BicD^24,25^. Thus, it is conceivable that a Nup358-min dimer binds to a BicD2 dimer, which is also in line with our ITC results, which reveal a single K_D_. While Nup358-min is a monomer on its own, it oligomerizes to form 2:2 complexes with either the BicD2-CTD or the KLC2, and also forms a ternary complex with both BicD2-CTD and KLC2 with 2:2:2 stoichiometry ^47,50^. We thus propose that cargo adaptor-induced dimerization may potentially be a universal feature of the activation mechanism, since BicD2 and KLC2 both form dimers in active dynein and kinesin-1 motors21,22,47,68.

– The random coil to helix transition would be better supported by CA & CB chemical shifts. With assignments in hand the chemical shifts for apo and bound could be graphed to highlight secondary structure propensity.

CA and CB chemical shifts would have been very useful. However, the Nup358-min NMR sample concentration can be no higher than 0.6 mM, above which the sample undergoes aggregation and is not stable. To obtain CA shifts in the bound state, we have attempted HN(CO)CA-CEST and due to the limited concentration of BicD2 sample, the S/N is too low for the data to be interpretable. However, the concept of coil to helix transition has already been well supported by both CEST and CD, in addition to mutagenesis data.

– Would be helpful to show the secondary structure prediction mentioned on page 14, what is the structure of the free protein, is there helical prediction or propensity in the region of the coil that transitions to helix?

We added the secondary structure prediction as a new panel (Figure S1d). Residues 2162-2184 are predicted by the PredictProtein Server to form an α-helix whereas the remainder of Nup358-min is predicted to be intrinsically disordered. However, based on experimental chemical shifts, there is no stable helical conformation in this region in the apo state.

– A discussion of what did we learn from the kinetics will be helpful. The kinetics of coil to helix are well reported in the literature. How are these similar?

Coil to helix transitions in model peptides are typically much faster in time scale than events characterized by the CEST method. We added these in the results:

“k_ex_ obtained here is much slower in time scale than the rate of coil-to-helix transition in model peptides. This suggests that the k_ex_ obtained by CEST include contributions from other events, such as binding and oligomerization, in addition to coil-to-helix transition.”

An appraisal of whether the authors achieved their aims, and whether the results support their conclusions.– Authors do appear to achieve their aims in determining the binding site on Nup358 for BicD2 while also proposing a mechanism for BicD2 cargo recognition– Suggested/implied conclusions leave the authors plenty of future work especially in regards to BicD2 regulation and possible bidirectional Nup complexes– The random coil to helix transition of Nup would be better supported by additional NMR and CEST data (mentioned above), however the coupling of CD data and secondary structure prediction (although not shown) do strengthen the conclusion– In the mutagenesis studies, particularly in Fig7a, the differences between some residues (2169, 2172, 2181) seem too small for the conclusions made.

Two criteria were used to select interface residues: the reduction of the ratio of bound BicD2-CTD/Nup358-min, normalized respective to the wild-type, was required to be greater than three times the error (which was calculated as the standard deviation from three independently performed experiments). Thus, all selected interface residues have a signal to noise ratio of 3 or more. In addition, for a residue to be an interface residue the binding ratio had to be lowered to at least 87% of the wild-type binding ratio. It should also be noted that residue 2169 and 2181 were identified as interface residues both by mutagenesis and in CEST, validating them further. In addition, residue 2172 was identified as an interface residue by CEST experiments. We decided not to select it as an interface residue for mutagenesis because the binding ratio was only 7% lower compared to the WT. However, the reduction of the binding ratio for residue 2172 is more than 3 times the standard deviation, so it is likely that this residue is an interface residue as well, given that it was identified by CEST too. It should be added that we have submitted our raw data to e*Life* and therefore readers can assess them. The following binding ratios were determined for alanine mutants of residues 2169, 2172 and 2181:

**Author response table 1. sa2table1:** 

	Binding ratio	Standard deviation	3x Standard deviation	Interface residue?
WT	1.00	n/a	n/a	n/a
2169	0.86	0.03	0.09	Yes
2172	0.93	0.01	0.03	No
2181	0.87	0.03	0.09	Yes

We do agree that the description of these cutoffs should be improved, and we have updated the figure legend to address this issue in Revised figure legend of Figure 7:

“The binding ratio was averaged from three independent experiments and the error was calculated as the standard deviation. Eleven interface residues were identified, which are colored red and show a reduction of the binding ratio upon mutation by at least three times the standard deviation. In addition, interface residues were required to have a binding ratio that was reduced by 13% or more compared to the wild-type.”

A discussion of the likely impact of the work on the field, and the utility of the methods and data to the community.

We added this point to the discussion.

“Our data thus suggest a structural basis for cargo recognition by BicD2 and will enable further studies aimed at identifying cell cycle specific regulatory mechanisms for distinct transport pathways that are facilitated by BicD2^4,11,50^. For example, our Nup358 mutants that selectively target the interaction with BicD2 can be used to dissect biological roles of the Nup358/BicD2 Rab6/BicD2 and nesprin-2G/BicD2 pathways in distinct stages of brain development.

It is very tempting to hypothesize that the cargo recognition helix of Nup358, intercalates into the coiled-coil of the BicD2-CTD to stabilize a coiled-coil registry shift in the Nup358/BicD2 complex, resulting in activation of BicD2 for dynein recruitment. This hypothesis remains to be experimentally confirmed.”

– Impact of the work is well summarized in the discussion/conclusion as these findings may be universal to many more dynein/cargo adaptor pairs– Methodology seems standard, although the future directions in regards to BicD2 regulation and possible bidirectional Nup complexes are interesting.– Any additional context you think would help readers interpret or understand the significance of the work.– Authors could discuss how/why this work applies to complexes that are not specific to dynein (other instances of autoinhibition regulation)

We added this point to the discussion. The principle of cargo-induced oligomerization could be a general mechanism for motor activation.

“Another intriguing possibility is that if kinesin binding dimerizes Nup358, this may be the key initial step leading to BicD2 activation and recruitment of dynein-dynactin to form a bidirectional complex (Figure 9). The ternary Nup358/BicD2/KLC2 complex may be a model for other transport modules, in which opposite polarity motors such as dynein and kinesin1 act together.”

– The repeated claim of a 20:1 (BicD2:Nup358) molar ratio for the CEST experiments is confusing, perhaps the authors mean 1:20? This should be corrected in multiple places.

We have fixed the reversal of the 20:1 and 1:20 Bic2:Nup358-min throughout the manuscript.

– The NMR titration is not a true titration as only a 1:1 molar ratio is shown. It is also unclear if peak intensities are taken from peak height or volume and a discussion of error bar calculation is needed to convince of the significance of the measurements.

We have addressed the point about not doing a full NMR titration. In addition to the 1:1 complex, we now show points for the 1:0.1 and 1:0.3 ratios used during the Nup358-Min:BicD2 titration in Figure S3b. The area of interest shows a consistent decline in intensity over that period. We used peak height, not volume (made clear in revised Figure 3 legend). Error bars are derived from the S/N of each peak determined from Sparky and propagated.

– To say that "Nup358 relieves the auto-inhibition of BicD2 so that dynein-dynactin can bind and move processivity on MTs" would perhaps be better stated as "functionally relieves auto-inhibition" since results are clearly not the same as when BicD2 is truncated and auto-inhibition is truly absent. (page 7)

This change was incorporated.

Reviewer #4:In this manuscript by Gibson and colleagues provides new insights into the structural and molecular basis for Nup358 binding to the dynein activating adaptor BicD2 and its implications in dynein's motility. In line with their previous publication (Noell et al., 2018), the authors show that the small fragment in Nup358 drives binding to BicD2. The authors quantitatively characterize the interaction using isothermal titration calorimetry (ITC) and confirm a binding stoichiometry of a Nup358 dimer to a BicD2 dimer. By using a combination of NMR spectroscopy, CD spectroscopy, pull-downs and SAXS experiments, they demonstrate that BicD2 recognizes Nup358 through a short "cargo recognition α-helix", which transitioned from a random coil to an α-helix upon BicD2 binding. Finally, the authors present evidence that the interaction between BicD2 and Nup358 is important for the formation of dynein/dynactin/BicD2/Nup358 motor complex, as well as the processive motility of the motor complex in vitro. Overall, the conclusions are well supported by the data and the findings clarify previously reported results. The solution NMR and in vitro motility data add significantly to the understanding of how cargo binding to activating adaptors may promote the processive motility of the motor complex.Pending points below, I think this manuscript merits publication in eLife.1. ITC experiments.a) In ITC experiments, the titration of protein (placed in the syringe) into the buffer (placed in the calorimetric cell) is lacking as a negative control. The control experiment is particularly important as the protein placed in the syringe is expected to be dimeric in solution (I assume Nup358 is placed in the syringe). The stepwise addition of the solution in the syringe into the cell might trigger the dissociation reaction of the Nup358 homodimer, leading to the detection of the heat.

We did perform the control titration of Nup358-min and GST-tagged Nup358-min (which was placed in the syringe) into the buffer (which was placed in the cell), and these controls were performed at the same time as the ITC titrations that are shown in the paper. These control titrations without BicD2-CTD resulted in a flat response characteristic of heat of dilution, which is not surprising, because apo Nup358-min is a monomer, which is confirmed by our previously published SEC-MALS analysis (Cui, *et al.*, 2019, Biochemistry 58: 5085). In the revised submission, we have added the control titrations as additional panels to Figure S1.

b) Please provide the experimental details on the ITC experiments.

We apologize for this oversight and thank the reviewer very much for this comment. The method section for the ITC experiments was added to the manuscript:

“Isothermal titration calorimetry (ITC)

ITC experiments were performed as previously described^50^. For ITC experiments, BicD2-CTD was placed in the cell of a calorimeter (GE Microcal ITC200) and titrated with Nup358-min either without the GST-tag (Figure 1) or with the GST-tag intact (Figure S1) at 25 °C. As a control, a second titration curve of Nup358-min (with or without the GST-tag) into buffer was recorded, which resulted in a flat line (Figure S1 a, b). The curve was fitted with the one-site model to determine the number of sites N, the equilibrium binding constant K, the change in entropy ∆S and the change in enthalpy ∆H. The affinity (dissociation constant Kd) was calculated as the inverse of K. Data was analyzed with the Origin software (OriginLab). For the titration of Nup358-min + BicD2CTD, the protein concentrations were 0.27 mM (Nup358-min) and 0.019 mM (BicD2CTD), respectively. For the titration of Nup358-min-GST + BicD2-CTD, the protein concentrations were 0.12 mM (Nup358-min-GST) and 0.013 mM (BicD2-CTD), respectively. The buffer contained 150 mM NaCl, 30 mM HEPES pH 7.5, 0.5 mM TCEP and 1 mM MgCl2.”

c) Please provide the concentrations of the proteins used in ITC experiments in the cell and the syringe.

The protein concentrations are found in the added methods section, see previous comment. For the titration of Nup358-min + BicD2-CTD, the protein concentrations were 0.27 mM (Nup358-min) and 0.019 mM (BicD2-CTD), respectively. For the titration of Nup358-min-GST + BicD2-CTD, the protein concentrations were 0.12 mM (Nup358-minGST) and 0.013 mM (BicD2-CTD), respectively.

2. NMR experiments.a) HSQC titration experiments (Figure 3 and Figure S2) are not very convincing as there are several mis-labeled (T2159 in Figure 3, A2440, T2143, G2146 in Figure S2 and so on) or unlabeled peaks. Also, the overall figure quality should be improved (figure resolution, tick marks, residue labels etc.).

We have fixed the incorrectly labeled residues (caused by transcription errors when moving from a 1-98 version of the Nup358-Min to the shifted by +2142 to match the whole complex), and also identified the G and L peaks that come from the linker region prior to the start of the homologous portion at D2148. We have also improved the overall quality of the figure, such as resolution and the sizes of tick marks and labels.

b) I highly suggest the authors perform at least one more titration experiment with a different molar ratio (e.g. [Nup358-min]:[BicD2-CTD]=1:0.5), to see whether the decrease of the peak intensities are truly dependent on BicD2 binding.

We have addressed the point about not doing a full NMR titration. In addition to the 1:1 complex, we now show points for the 1:0.1 and 1:0.3 ratios used during the Nup358-Min:BicD2 titration in Figure S3b (also shown below). The area of interest shows a consistent decline in intensity over that period. We used peak height, not volume. Error bars are derived from the noise of the NMR spectra in the region where no signal is present.

c) It would be nice if the authors analyze the secondary structure elements using NMR chemical shifts (using TALOS+ for example) and compare the results with the helical propensity detected by CD experiments.

From Talos analysis (see figure below), there is no clear helical propensity for Apo Nup358-min, in agreement with CD experiments.

d) The assignment of apo Nup358-min should be deposited in the BMRB database.

This is being carried out and the BMRB ascension number will be provided during the revision of this MS.

3. Single molecule binding and processivity assays.a) It is surprising that DDBNmin-zip complex formed with the Nup358min-zip-L2177A behaves similarly to WT-Nup358min-zip. Mutagenesis of L2177A impairs the interaction between Nup358 and BicD2 (Figure S12a), and L2177 residue lies at the center of the predicted binding interface (Figure 7c). Could the authors comment on this?

It is indeed surprising that L2177A behaves similarly as WT in TIRF measurements, which is not fully understood. We speculate that there is some plasticity at the Nup358/BicD2 interface, as we have stated in the MS. However, four of the five characterized single amino acid mutants were able to disrupt the progressive motility in DDBN complex, indicating that Nup358/BicD2 Interfaces in the minimal Nup358/BicD2 complex and the intact DDBN complex are similar.

[Editors’ note: what follows is the authors’ response to the second round of review.]

Essential revisions:1) Perform replicates of the single molecule experiments showing activation of BICD2 by Nup358 and make the raw data available (please see reviewer 1)

Figure 2f now shows data from 3 replicate experiments for each condition and each time DDBN displayed higher motility than DDB. These experiments confirm that the Nup358-BicD2 interaction activates BICD2 for processive motion of the dynein-dynactin complex. As requested by reviewer 1, we have provided 12 representative movies as Figure 2 supplements for DDBN, DDB, DDB^CC1^, and DDN (3 sets of data for each condition). In addition, we presented one set of 3 movies for DDBN, DDB and DDB^CC1^ in the main paper. More data are presented for “the number of processively moving complexes” in Figure 2f. We have also added more data to the speed and run length histograms in Figure 2g and 2h.

In addition, we have analyzed more data for the mutant L2177A which showed similar run frequency, speed and run length as WT-Nup358 in the original MS. After adding additional data for both DDBN-WT and DDBN-L2177A, the speed and run length of DDBN-L2177A were shown to be significantly different from the WT complex as were the other mutants, although the run frequency is still similar to the WT (with p value 0.09). Because mutant DDBN-L2177A shows reduced speed and run length and is not an outlier, we merged the relevant SI on L2177A in the original MS into Figure 8.

2) Tone down some of the conclusions from the NMR work (reviewer 2)

We have modified the abstract to emphasize that the coil-to-a-helix transition is also supported by circular dichroism spectroscopy. We have also changed the section title for CEST from “CEST revealed a coil-to-a-helix transition at BicD2-Nup358 interface” to “CEST suggests a coil-to-a-helix transition at the BicD2-Nup358 interface”. Furthermore, we added the caveats mentioned by reviewer 2 to the discussion:

Edited discussion about CEST NMR:

“The NMR titration of the BicD2-CTD into ^15^N labeled Nup358-min narrowed the binding region to the N-terminal half of Nup358-min. Furthermore, we obtained chemical shifts for the CA and CB atoms for the apo state of Nup358-min, which confirm that the apo-state is intrinsically disordered (Figure 3 —figure supplement 3). Due to slow chemical exchange on the NMR time scale and fast relaxation in the bound state, the BicD2-bound state of Nup358-min cannot be directly observed in NMR, and thus a powerful solution NMR technique, CEST, was applied to map the chemical shifts of the “invisible state”. This approach not only identified key residues at the Nup358/BicD2 interface, but also showed that chemical shift changes upon binding correspond to that of a coil-to-a-helix transition in Nup358-min. Notably, a conformational change from coil-to-a-helix was consistent with our results from CD spectroscopy and SAXS, and the BicD2/Nup358 interface was validated by mutagenesis. The observed time scale is also in line with other proteins undergoing coil-to-a-helix transitions (Charlier et al.). However, it should be noted, that in order to obtain higher-resolution structural information, the chemical shifts for the CA and CB atoms of Nup358-min in the bound state remain to be determined in the future. Notably, our functional assays highlight the important role of the interaction between Nup358 and BicD2 in dynein activation. Single molecule binding assays showed that mutagenesis of the Nup358 cargo-recognition a-helix diminished the interaction of Nup358-min with full-length BicD2 and decreased the formation of dynein/dynactin/BicD2/Nup358 complexes (DDBN). Furthermore, the mutant DDBN complexes that were formed exhibited reduced run length and speed in single-molecule processivity assays on microtubules reflecting formation of a less stable complex. Our data highlight the important role of dynein-activating-adaptor/adaptor-binding protein interactions in activating and fine-tuning dynein motility.”

Reviewer #1:1) Regarding Figure 2f, did the authors perform 3 replicates for each condition? This is not specified in the text and is not deducible from the source data provided. To make the claim that Nup358 activates DDB, it is necessary to show that DDBN consistently displays higher motility than DDB in 3 replicates (for each condition). Also, the source data provided by the authors for this experiment is somewhat unhelpful to the reviewers. For the sake of transparency, the authors should provide access to the source movies for this experiment.

Figure 2f now shows data from 3 total experiments for each condition and each time DDBN displayed higher motility than DDB. These experiments confirm that the Nup358-BicD2 interaction activates BICD2 for processive motion of the dynein-dynactin complex. As requested by reviewer 1, we have provided 12 representative movies in source data for DDBN, DDB, DDB^CC1^, and DDN (3 sets of data for each condition). In addition, we presented one set of 3 movies for DDBN, DDB and DDB^CC1^ in the main paper.

2) The second sentence in the abstract "Nup358 binds and activates the auto-inhibited dynein adaptor Bicaudal D2 (BicD2), which in turn recruits and activates the dynein machinery to position the nucleus" implies that activation of BICD2 by Nup358 had been shown previously. To my knowledge, direct activation of BICD2 by Nup358 had not been formally shown before (although we knew that Nup358 recruits dynein via BICD2). The ability of Nup358 to relieve BICD2 from its autoinhibited state in in vitro assays is one of the key discoveries of this manuscript and should be emphasized in the abstract/introduction.

We thank the reviewer for this great suggestion. We have modified the abstract to emphasize this:

“Nup358 interacts with the dynein adaptor Bicaudal D2 (BicD2), which in turn recruits the dynein machinery to position the nucleus. However, the molecular details of the Nup358/BicD2 interaction and the activation of transport remain poorly understood. Here for the first time, we show that a minimal Nup358 domain activates dynein/dynactin/BicD2 for processive motility on microtubules.”

We changed the last paragraph of the introduction to contain the following sentences:

“Direct activation of BicD2 by Nup358 has not been shown before. Here, we use single-molecule binding and processivity assays to show that a minimal dimerized Nup358 construct is sufficient to activate full-length dynein/dynactin/BicD2 complexes for processive motility.”

3) The introduction is poorly written and could be improved.– The first paragraph of the introduction could be rearranged to better emphasize what is known (i.e. that cargo binding to the CTD frees up the NTD for interaction with dynein/dynactin) and what is unknown (i.e. what are the structural basis of cargo recognition and how does this lead to structural changes in BICD2 conformation). For example, the authors first state that "Cargoes are required to activate BicD2 for dynein binding, which is a key regulatory step for dynein-dependent transport but the underlying mechanism is unknown" but later provide a detailed explanation of how BICD2 is autoinhibited "in the absence of cargoes, BicD2 assumes a looped, auto-inhibited conformation, in which its N- terminal dynein/dynactin binding site binds to the CTD and remains inaccessible. The CTD is required for auto-inhibition, as a truncated BicD2 without the CTD activates dynein/dynactin for processive motility. Binding of dynein adaptors/cargo to the CTD releases auto-inhibition, likely resulting in an extended conformation", which is somewhat confusing.

We thank the reviewer for this suggestion. We have deleted the half sentence “but the underlying mechanism is unknown”; and edited the last sentence of first paragraph in the introduction accordingly:

“However, to date, there are no detailed structural studies of a BicD2/cargo complex and the structural mechanisms of BicD2-mediated cargo recognition and dynein activation still remain poorly understood.”

– The sentence "Finally, Rab6GTP recruits BicD2/dynein for the transport of Golgi-derived secretory vesicles" does not fit well with the rest of the paragraph in its current form. Something along the lines "Apart from its roles in nuclear positioning, BicD2 is also involved in the transport of Golgi-derived secretory vesicles. In this process, BicD2 and dynein are recruited by the vesicle-associated small GTPase Rab6" would be more helpful to the reader.

We made this change as suggested.

– The paragraph on the importance of IDRs is not very informative, since the idea of an IDR that transitions into a cargo-recognition helix has not been introduced at this stage. In my opinion, this paragraph could be removed and replaced by a short sentence in the discussion which highlights the importance of IDRs in dynein biology. If the authors wish to keep it, they should provide more context on why IDRs are relevant to the scope of this research.

We edited the IDR paragraph in the introduction to provide more context:

“IDRs play important roles in dynein biology. For example, an intrinsically disordered region in the dynein light intermediate chain 1 (LIC1) undergoes a coil-to helix transition when it interacts with the N-terminal domain of BicD2, which is an important step in the activation of dynein for processive motility^40,41,43^.”

Reviewer #2:The discussion of the ITC fits was improved and is clear now.The explanation for the absence of CA and CB chemical shifts is not, however. A concentration of 0.6 mM should be sufficient for assignments of the free protein. It is not clear to me what experimental chemical shift the author is referring to that do not show stable helical conformation. Were these published before, and if so they will be better included here in the figure instead of prediction. A nascent helix will be sufficient to show.

To address this comment, we edited the text to tone down the conclusions from the NMR spectroscopy. See our response to the editor’s comments above.

Furthermore, we wanted to clarify that CA and CB shifts are not available for the BicD2-bound state of Nup358-min through CEST experiments. Indeed, the CA and CB shifts are available for the apo-state of Nup358-min through the regular HNCACB types of experiments. We have now deposited CA and CB shifts of the apo state of Nup358-min BMRB (accession # 51282), which has not been published before. Using Talos, a highly accurate secondary structure prediction software developed by Ad Bax’s group, we show that there is no a-helical content or secondary structure propensity based on these chemical shifts in the apo state (Figure 3-supplementary figure 3). Secondary structure predictions from the protein sequence are often not accurate. Talos data, in addition to our CD data (Figure 5), clearly demonstrate that Nup358-min adopts a random coil conformation in the apo-state.

Also, it is concerning to me that the timescale measured by CEST is so different from coil to helix transition, but still the interpretation of coil to helix transition as the driver has not changed. If CEST is measuring a different process than coil to helix, and there is no strong structural evidence that there is a helix being formed or has tendency to form, then the interpretation of coil to helix transition is not supported.

To address this comment, we edited the text to tone down the conclusions from the NMR spectroscopy. See our response to the editor’s comments.

The most important information from CEST is the chemical shifts of BicD2-CTD-bound Nup358-min (the “invisible” state), which strongly suggests a coil-to-helix transition from apo to BicD2-CTD-bound state. It should be emphasized that the coil-to-a-helix transition is supported not only by the chemical shifts of CO and N of the bound state but also by CD, SAXS measurements and mutagenesis.

The interpretation of k_ex_ is more complicated. Coil to a-helix transitions in model peptides are typically much faster in time scale than events characterized by the CEST method, as shown by Charlier et al. This is because k_ex_ obtained by CEST include contributions from other events, such as binding and oligomerization, in addition to coil-to-a-helix transition.